# The replicative helicase CMG is required for the divergence of cell fates during asymmetric cell division in vivo

Nadin Memar [1,4] ✉, Ryan Sherrard[2,5], Aditya Sethi [1,5], Carla Lloret Fernandez [1,5], Henning Schmidt[3], Eric J. Lambie [1], Richard J. Poole [1], Ralf Schnabel[3] & Barbara Conradt [1] ✉

We report that the eukaryotic replicative helicase CMG (Cdc45-MCM-GINS) is required for differential gene expression in cells produced by asymmetric cell divisions in *C. elegans*. We found that the *C. elegans* CMG component, PSF-2 GINS2, is necessary for transcriptional upregulation of the pro-apoptotic gene *egl-1* BH3-only that occurs in cells programmed to die after they are produced through asymmetric cell divisions. We propose that CMG's histone chaperone activity causes epigenetic changes at the *egl-1* locus during replication in mother cells, and that these changes are required for *egl-1* upregulation in cells programmed to die. We find that PSF-2 is also required for the divergence of other cell fates during *C. elegans* development, suggesting that this function is not unique to *egl-1* expression. Our work uncovers an unexpected role of CMG in cell fate decisions and an intrinsic mechanism for gene expression plasticity in the context of asymmetric cell division.

The ability of cells to adopt different fates is fundamentally important to life. One process through which cells of different fates are generated is through asymmetric cell division. The mechanisms through which gene expression patterns, and hence cell fates, diverge during asymmetric cell division are incompletely understood[1–4]. A large body of work has demonstrated that different daughter cell fates can be established through the asymmetric inheritance of cell fate determinants in the form of protein or mRNA. Whether or not epigenetically distinct chromosomal loci also act as asymmetrically inherited cell fate determinants is not clear. The quantal cell cycle theory proposes that changes during chromosome replication make regions of the genome available for transcription in daughter cells that were not available for transcription in the mother cell[5,6]. Whether and how this may occur in vivo is unknown.

The development of the nematode *C. elegans* is essentially invariant and provides a unique opportunity to study how cells acquire specific fates during their lineage history, i.e., the successive rounds of cell division starting from the first division of the one-cell embryo to the terminal division that generates that cell[7]. Most terminal divisions during *C. elegans* development are asymmetric and result in two daughter cells that adopt different fates[8,9]. Lineage-resolved single-cell transcriptome profiling of *C. elegans* embryos has revealed that in the case of such terminal asymmetric divisions, the genes that determine the two daughter cell fates are often co-expressed in the mother cell[10], a phenomenon observed in other organisms and referred to as multilineage priming[11–17]. The intrinsic and extrinsic mechanisms through which the expression of genes subject to multilineage priming is modified during terminal asymmetric divisions to retain expression in one specific daughter cell are not fully understood.

Out of the 1090 somatic cells formed during the development of a *C. elegans* hermaphrodite, 131 reproducibly die[8,9]. Cell death during *C. elegans* development can therefore be considered a genetically programmed cell fate. Most cells that adopt the cell death fate are generated through an asymmetric cell division and after the completion of

[1]Research Department Cell and Developmental Biology, Division of Biosciences, University College London, London, UK. [2]Faculty of Biology, Ludwig-Maximilians-University Munich, Planegg-Martinsried, Germany. [3]Institute of Genetics, TU Braunschweig, Braunschweig, Germany. [4]Present address: Center for Genomic Integrity, Institute for Basic Science (IBS), Ulsan, South Korea. [5]These authors contributed equally: Ryan Sherrard, Aditya Sethi, Carla Lloret Fernandez. ✉e-mail: nmemar@ibs.re.kr; b.conradt@ucl.ac.uk

mother cell division, very rapidly (within 20–30 min) undergo apoptotic cell death[18,19]. The gene that determines the fate of these cells is *egl-1*, which encodes a BH3-only protein, a pro-apoptotic member of the Bcl-2 superfamily of cell death regulators[20–22]. *egl-1* is necessary and sufficient for apoptotic cell death, and its expression is essentially restricted to lineages in which an apoptotic cell death occurs. Using single-molecule RNA Fluorescence In Situ Hybridization (smRNA FISH), the dynamics of *egl-1* mRNA levels has been analyzed in specific cell death lineages in vivo[23]. This revealed that a low level of *egl-1* mRNA is already present in the mother cell. Immediately after mother cell division, the *egl-1* mRNA level increases substantially in the daughter that dies but decreases to zero in the daughter that survives. This suggests that mothers of cells that die are "poised" for *egl-1* expression and that the regulation of *egl-1* expression is binarized during the terminal asymmetric division to result in increased *egl-1* expression specifically in the daughter that dies. Indeed, lineage-resolved single cell transcriptome profiling of *C. elegans* embryos identified *egl-1* as a gene that is subject to multilineage priming[10], confirming the smRNA FISH results[23]. It has been proposed that the non-random segregation of direct repressors of *egl-1* transcription into the daughters that survive contributes to repression of *egl-1* expression in these cells[24,25]. How *egl-1* expression is increased in the dying daughters has so far been unclear.

Here we demonstrate that the increase in *egl-1* BH3-only expression in the daughters that die is dependent on the eukaryotic replicative helicase CMG (Cdc45-MCM2-7-GINS), and we provide evidence that this requirement of CMG is separable from its role in DNA replication. In addition, we demonstrate that the role of CMG in the divergence of cell fates during asymmetric cell division is not restricted to the cell death fate. Our results uncover an intrinsic mechanism through which the expression of a gene that is subject to multilineage priming can be altered during terminal asymmetric cell divisions. Importantly, they also provide in vivo evidence for the role of components of the replisome in the control of gene expression during asymmetric cell division, thus providing experimental support for the quantal cell cycle theory.

## Results

### Role of *psf-2* GINS2 in embryonic development

We identified the *t3443*ts mutation by screening a collection of temperature-sensitive (ts) embryonic lethal mutants for abnormalities in the invariant pattern of cell death. At the non-permissive temperature (25 °C), the morphology of early *t3443*ts embryos is essentially indistinguishable from that of wild-type embryos (Fig. 1A, 4-cell stage, Pre-morphogenetic stage [-350-cell stage]) (see Methods for the exact time of the shift from permissive to non-permissive temperature). The different tissues can be recognized, and a normal pre-morphogenetic stage is reached, which suggests that tissue differentiation overall is not impaired. However, *t3443*ts mutants undergo arrest shortly after the initiation of morphogenesis (Fig. 1A, see "Final recording"). This embryonic lethal (Emb) phenotype is fully penetrant (100% embryonic lethal) (Fig. 1B). At the permissive temperature (15 °C), *t3443*ts animals are viable (Fig. 1B). Furthermore, there is no significant difference in brood size (number of eggs laid) between *t3443*ts and wild type at the permissive or non-permissive temperature (Fig. 1B)

Using snip-SNP mapping[26], we mapped the Emb phenotype of *t3443*ts animals to Linkage Group I (LGI) close to the variation pkP1071 at position 23.40 cM and performed whole genome sequencing. In the region identified, we found one gene that carries a missense mutation in its coding region in *t3443*ts animals, the gene *psf-2* (yeast Partner of Sld Five). *psf-2* encodes the *C. elegans* ortholog of Psf2, one of four subunits (Psf1, Psf2, Psf3, Sld5) of the GINS (Go-Ichi-Ni-San) complex[27]. (Of note, Psf1, Psf2, Psf3 and Sld5 are also referred to as GINS1, GINS2, GINS3 and GINS4, respectively.) GINS is a subcomplex of the conserved replicative helicase CMG, which unwinds double-stranded DNA prior to DNA synthesis and is therefore essential for DNA replication[28,29]. *t3443*ts is a missense mutation that causes a cytosine to thymine change at position 190 (C190T) of the coding region of the *psf-2* gene (Suppl Fig. 1A). This results in a predicted proline to serine change at position 64 of the amino acid sequence of the PSF-2 protein (P64S), a residue that is conserved from *S. cerevisiae* to human[27] (Suppl Fig. 1B, C).

To confirm that *t3443*ts is an allele of *psf-2*, we amplified a 3.8 kb genomic fragment, which spans the *psf-2* transcription unit and upstream and downstream regions (Suppl Fig. 1A, B *bcEx1302* 3.8 kb) from wild type and injected it into *t3443*ts animals to generate a *psf-2*(+) transgene. We found that this transgene rescues the Emb phenotype of *t3443*ts animals. *t3443*ts embryos carrying the *psf-2*(+) transgene complete embryogenesis and hatch (Fig. 1A–C, *psf-2(t3443*ts*)*; (*psf-2*(+)). As an additional form of verification, we knocked-down the *psf-2* gene by RNA interference (RNAi) in wild-type animals for 24 h (see Fig. 1A and Suppl Fig. 2 for phenotype at final recording). (*psf-2(RNAi)* for 24 h results in a weak loss-of-function phenotype.) We found that this causes an Emb phenotype in the F1 progeny that resembles the phenotype observed in *psf-2(t3443*ts*)* animals at the non-permissive temperature (Fig. 1A, *psf-2(RNAi)*). These results demonstrate that *t3443*ts is a loss-of-function mutation of the gene *psf-2*, which is verified further by the experiments described below.

To analyze the development of *psf-2(t3443*ts*)* animals in more detail, we performed 4D microscopy using Differential Interference Contrast (DIC) combined with cell lineage analyses ("4D lineaging")[30,31]. We allowed one-cell embryos to undergo four rounds of cell division, identified the ABarp blastomere and measured cell cycle length during the five consecutive rounds of cell division that give rise to the cell ABarpppppp, which differentiates into the hypodermal cell V6R (Fig. 1C, D). ABarpppppp is one of the few cells that differentiates after the 9th round of cell division. For this reason, it serves as a control for 1st wave cell deaths, which occur after the 9th round of cell division. In wild-type animals, cell cycle lengths increase from an average of 22 min to an average of 40 min during these five rounds of cell divisions. In *psf-2(t3443*ts*)* animals, during the same five rounds of cell division, cell cycle lengths increase from an average of 39 min to an average of 144 min (Fig. 1C, D). Therefore, cell cycle length in *psf-2(t3443*ts*)* animals is increased almost twofold at the beginning of the recordings (22 min versus 39 min) and almost 4-fold at the end of the recording (40 min versus 144 min). This "increased cell cycle length" phenotype of *psf-2(t3443*ts*)* animals is rescued by the *psf-2*(+) transgene (Fig. 1C, D). Furthermore, the knock-down by RNAi of *psf-2* (for 24 h) increases cell cycle length similarly to what we observed in *psf-2(t3443*ts*)* animals at the non-permissive temperature (Fig. 1C, D). *psf-2(RNAi)* also leads to the block of some cell divisions. For example, in the cell lineages shown in Fig. 1D, the divisions of the cells ABarpppaa and ABarpppap were blocked in *psf-2(RNAi)* animals (indicated in red). Finally, the knock-down of *psf-2* by RNAi for 48 h leads to an arrest at about the 50-cell stage (see Suppl Fig. 2 for phenotype at final recording). (*psf-2(RNAi)* for 48 h causes a strong loss-of-function phenotype.) To confirm that the increased cell cycle length phenotype observed in *psf-2(t3443*ts*)* and *psf-2*(RNAi) animals is not specific to the ABarpppppp lineage, we also analyzed the MSpppppp lineage. We identified the MS blastomere at the 16-cell stage and measured cell cycle lengths during the six consecutive rounds of cell division that give rise to the cell MSpppppp. As in the ABarpppppp lineage, we found an increased cell cycle length phenotype in *psf-2(t3443*ts*)* animals at the non-permissive temperature, and this phenotype was rescued by *psf-2*(+) (Suppl Fig. 3A, B). In summary, in line with the essential role of CMG in DNA replication, reducing *psf-2* GINS2 function causes increased cell cycle lengths, ultimately resulting in a block in cell division and embryonic lethality.

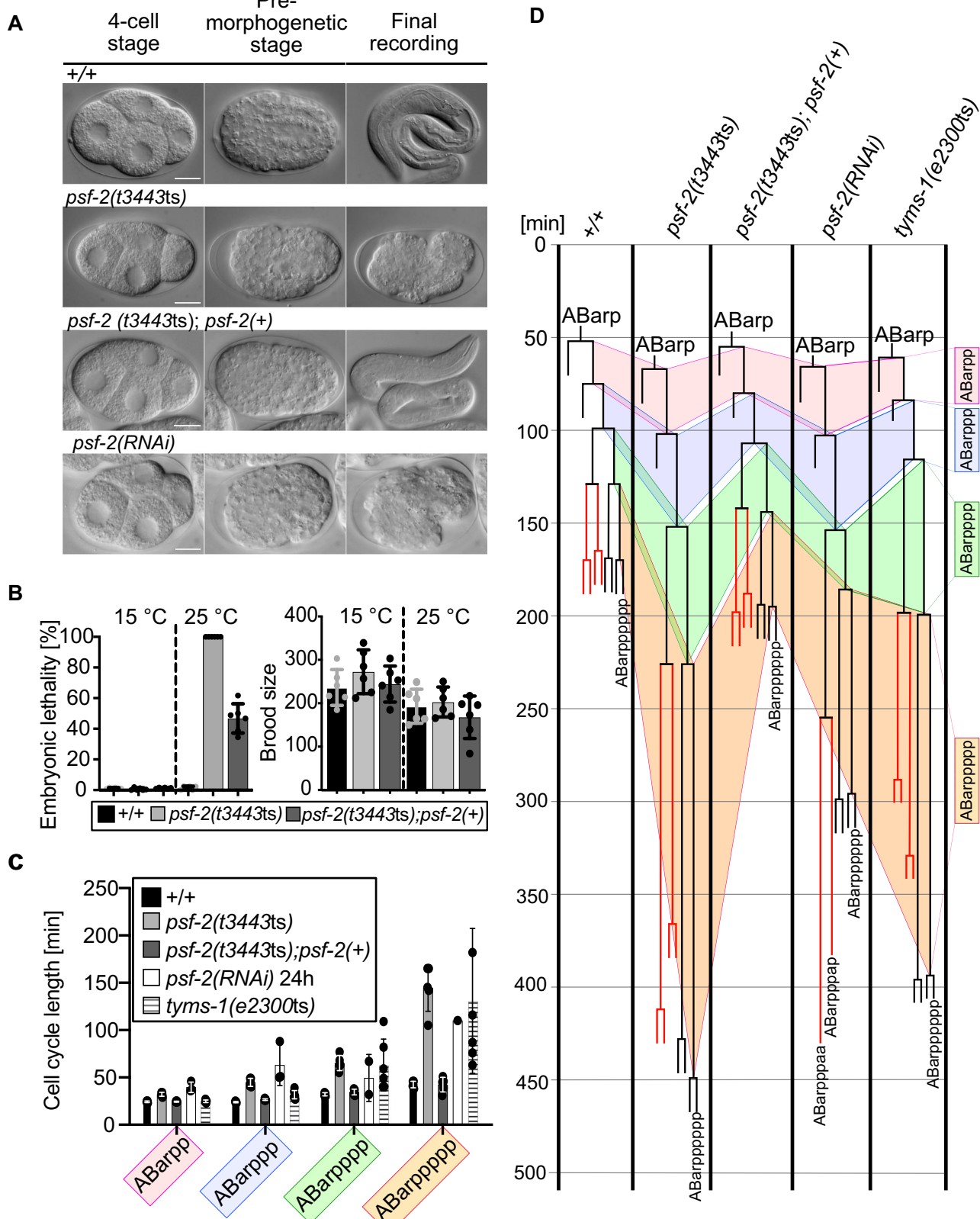

## Role of *psf-2* GINS2 in cell death

During *C. elegans* development, 131 cells adopt the cell death fate and reproducibly die[8,9]. Using the same 4D recordings as above, we analyzed the number of cell deaths in *psf-2*(*t3443*ts) embryos at the non-permissive temperature to determine whether the cell death fate was affected in these mutants. After the 8th round of cell division, one cell

derived from the MS blastomere dies (MSpaapp) and after the 9th round of division, 13 cells derived from the AB blastomere die, together referred to as the 1st wave of cell death. We identified the 13 cell death lineages derived from AB and determined the fate of the 13 cells that normally die. As shown in Fig. 2A, we found that in wild type, all 13 cells die (+/+, 0% cell death blocked, *n* = 4). In contrast, in *ced-3*(*n717*)

**Fig. 1 | Reducing *psf-2* GINS2 function causes increased cell cycle lengths, a block in cell division and embryonic lethality. A** Differential Interference Contrast (DIC) images of representative wild-type (+/+), *psf-2(t3443*ts), *psf-2(t3443*ts); *psf-2*(+) (transgene *bcEx1302*) and *psf-2(RNAi)* embryos at the 4-cell stage, the pre-morphogenetic stage and at the final recording (terminal phenotype). The +/+ and *psf-2(t3443*ts); *psf-2*(+) embryos completed embryogenesis and hatched. Representative images were taken from long-term recordings performed at 25 °C. Scale bars 10 µm. **B** Embryonic lethality [%] and Brood size at permissive (15°C) and non-permissive temperature (25 °C) in wild-type (+/+), *psf-2(t3443*ts) and *psf-2(t3443*ts); *psf-2*(+) (transgene *bcEx1302*) animals. For each genotype, embryonic lethality and

brood size of the entire progeny of six adults was analyzed (*n* = 6). Mean ± SD are indicated. Data plots here and throughout were done using GraphPad Prism 10. **C** Cell cycle length [min] of the ABarppppp cell and its ancestors in wild-type (+/+) (*n* = 5), *psf-2(t3443*ts) (*n* = 5), *psf-2(t3443*ts); *psf-2*(+) (transgene *bcEx1302*) (*n* = 5), *psf-2(RNAi)* (*n* = 3) and *tyms-1(e2300*ts) (*n* = 6) animals at 25 °C. Mean ± SD are indicated. **D** ABarppppppp (V6R) lineage of representative animals of the genotypes indicated. The wild-type and *psf-2(t3443*ts); *psf-2*(+) embryos completed embryogenesis and hatched. The blocks in cell division of ABarpppaa and ABarpppap in the *psf-2(RNAi)* embryo and the corresponding lineages in the other genotypes are indicated in red.

---

animals, which exhibit a block in cell death or Ced phenotype[32], 100% of cell death is blocked (a cell death was considered blocked, when the cell had not turned into a cell corpse based on DIC after twice the cell cycle length of its mother cell; see Methods) (Fig. 2B). We analyzed four *psf-2(t3443*ts) embryos and found that at the non-permissive temperature, the death of many of the 13 cells is blocked (Fig. 2A, *psf-2(t3443*ts)). In total, we found that at the non-permissive temperature, 63% of the 1st wave cell deaths derived from AB are blocked in *psf-2(t3443*ts) animals (Fig. 2B). In addition, we found that the death of MSpaapp is blocked in four out of six *psf-2(t3443*ts) embryos analyzed (67% cell death blocked) (Fig. 2C). Of note, *psf-2(t3443*ts) can affect each of the 1st wave cell deaths (Fig. 2A), which indicates that it does not cause a lineage-specific block in cell death. Importantly, the *psf-2*(+) transgene fully rescues the Ced phenotype observed in *psf-2(t3443*ts) animals (Fig. 2A-C). In addition, the knock-down of *psf-2* by RNAi (for 24 h) blocks 64% of the 1st wave cell deaths derived from AB (Fig. 2A, B). This confirms that the Ced phenotype observed in *t3443*ts animals is caused by a reduction in *psf-2* GINS2 function. For the 1st wave cell deaths derived from AB that were not blocked in *psf-2(t3443*ts) animals (i.e., 37% of the cell deaths), we measured the time it took the cells to die. Specifically, we measured the time between the completion of the mother cell division that gives rise to a particular cell and that cell's adoption of a cell corpse appearance by DIC. We found that in wild type, it takes a cell on average 23.6 min to die (Fig. 2D). In contrast, in *psf-2(t3443*ts) animals, it takes a cell on average 49.7 min to die (Fig. 2D). This indicates that in the 1st wave cell deaths that still occur in *psf-2(t3443*ts) animals (37% of all 1st wave cell deaths analyzed), cell death is delayed. Finally, the 1st wave cell deaths derived from the AB blastomere are generated through asymmetric cell divisions. To rule out that *psf-2(t3443*ts) causes a switch of daughter cell fates rather than a block in cell death, we analyzed the fates of the sisters of the 13 AB-derived 1st wave cell deaths in four *psf-2(t3443*ts) embryos (Suppl. Fig. 4). In wild type, 100% of the sister cells divide and give rise to two daughter cells. In *psf-2(t3443*ts) animals, 65.8% of the sister cells divide and give rise to two daughter cells. In the remaining 34.2%, cell division is blocked (Suppl. Fig. 4A, B). Importantly, 0% of the sister cells inappropriately die (Suppl. Fig. 4B), confirming that *psf-2(t3443*ts) does not cause a switch of daughter cell fates. These results are exemplified in Supplementary Fig. 4C for the sister of the 1st wave cell death ABalaapapa referred to as CD#1. In three out of four *psf-2(t3443*ts) embryos (embryos #1-3), the sister of CD#1 (ABalaapapp) divides and in the remaining embryo (embryo #4), it fails to divide.

Of the 131 cell deaths that occur during *C. elegans* development, 18 occur post-embryonically[8]. To determine whether *psf-2(t3443*ts) also blocks post-embryonic cell deaths, we analyzed the death of the cell QL.pp, which dies in larvae of the 1st larval stage (L1 larvae). Using a Q lineage specific reporter (P*toe-2gfp*)[33], we found that at the non-permissive temperature, QL.pp was blocked in 29% of *psf-2(t3443*ts) animals (Fig. 2E).

Based on these results we conclude that reducing *psf-2* GINS2 function not only causes a fully penetrant Emb phenotype, but also an incompletely penetrant Ced phenotype. In addition, the Ced phenotype exhibits variable expressivity, ranging from a block in cell death to

an increase in the time it takes a cell to die. Hence, *C. elegans psf-2* GINS2 is required for the cell death fate.

### Role of CMG in cell death

PSF-2 is a component of the GINS complex, which comprises four proteins, PSF-1, PSF-2, PSF-3 and SLD-5[27]. To determine whether the Ced phenotype detected in *psf-2(t3443*ts) animals can be attributed to the loss of GINS subcomplex function rather than the specific reduction of *psf-2* GINS2, we used RNAi to knock-down expression of the gene *psf-3* and found that 60% of the 1st wave cell deaths derived from AB are blocked (Fig. 2B, *psf-3(RNAi)*) (see Methods for the exact timing and duration of the RNAi knock downs).

The assembly of the replicative helicase CMG at replication origins occurs in a stepwise process[28,29]. After mitosis, a hexameric ring of the ATPases MCM2-7 is assembled around double-strand DNA at replication origins where it forms part of the pre-replication complex. CMG assembly is completed in S phase when CDC45 and the pre-formed GINS complex are recruited to MCM2-7 rings[28,29]. The loss of *psf-2* or *psf-3* is predicted to disrupt the formation of the GINS complex and the assembly of CMG during S phase but not the assembly of MCM2-7 rings at replication origins after mitosis. To determine whether the Ced phenotype detected in *psf-2(t3443*ts) and *psf-3(RNAi)* animals can be attributed to a more general loss of CMG function, we knocked-down the genes *mcm-2* and *mcm-7*[34] and found that 58% and 54% of the 1st wave cell deaths derived from AB are blocked, respectively (Fig. 2B, *mcm-7(RNAi)*, *mcm-2(RNAi)*). These results demonstrate that several members of the CMG complex are required for the cell death fate and suggest that the Ced phenotype is the result of the inability to assemble CMG in S phase.

### Relationship between cell cycle length and cell death

Reducing *psf-2* GINS2 function causes two phenotypes, an Emb phenotype, which we have shown above is the result of increased cell cycle lengths (and ultimately a block in cell division) likely caused by replication defects, and a Ced phenotype. To address whether there is a causal relationship between these two phenotypes, we measured the cell cycle lengths of the mothers of the 1st wave cell deaths derived from the AB lineage in the wild-type and *psf-2(t3443*ts*)* embryos that we had analyzed for a block in cell death (Fig. 3A; and see data in Fig. 2A, B). We found that in wild type, in which 0% of the cell deaths are blocked, the average cell cycle length of mothers is 42 min (Fig. 3B). In *psf-2(t3443*ts) animals, in which 63% of the cell deaths are blocked, there are two groups of mother cells: (1) mothers, whose daughters die ("cell death") (37% of the mothers) and (2) mothers, whose daughters fail to die ("cell death blocked") (63% of the mothers). As seen in Fig. 3B (*psf-2(t3443*ts*)*), in both groups of mothers, there is a broad range of cell cycle lengths, reflecting the increased cell cycle length caused by *psf-2(t3443*ts*)*. The average cell cycle lengths of the two groups are 108 min and 101 min, respectively, which is significantly different from the average cell cycle length observed in wild type (+/+); however, there is no significant difference between the average cell cycle lengths of these two groups (Fig. 3B). Furthermore, in both groups (i.e., regardless of whether their daughters died or not), there are mothers

**A**

| Cell | +/+ | psf-2(t3443ts) #1 | #2 | #3 | #4 | psf-2(t3443ts);psf-2(+) #1 | #2 | #3 | #4 | psf-2(RNAi) #1 | #2 | #3 |
|---|---|---|---|---|---|---|---|---|---|---|---|---|
| ABalaapapa | | | | | | | | | | | | |
| ABalaappaa | | | | | | | | | | | | |
| ABalapapaa | | | | | | | | | | ? | | ? |
| ABalappaaa | | | | | | | | ? | ? | | | |
| ABalppaaaa | | | | | | | | | | | | |
| ABalppaapa | | | | | | | | | | | | |
| ABaraaaapp | | | | | ? | | | ? | | | | |
| ABarpaaapp | | | | | ? | | | | | | ? | |
| ABplpappap | | | | | | | | | | | | |
| ABplppaaap | | ? | ? | ? | ? | | | | | ? | | ? |
| ABplpppapp | | | | | | | | | | | | |
| ABprppaaap | | | ? | | ? | | | | | ? | ? | ? |
| ABprpppapp | | | | | | | | | | | | |

Legend: ⬤ cell death | ◯ cell death blocked | ⊁ cell division of mother blocked | ⊥ mother dies | ? cell lost

**B**

| Genotype | % cell deaths blocked during first wave | n |
|---|---|---|
| +/+ | 0 | 52 |
| ced-3(n717) | 100 | 32 |
| psf-2(t3443ts) | 63 | 41 |
| psf-2(t3443ts) 15°C | 0 | 32 |
| psf-2(t3443ts);psf-2(+) | 0 | 51 |
| psf-2(RNAi) | 64 | 28 |
| psf-3(RNAi) | 60 | 25 |
| mcm-7(RNAi) | 54 | 26 |
| mcm-2(RNAi) | 58 | 19 |
| tyms-1(e2300ts) | 0 | 31 |

**C**

| Genotype | % MSpaapp cell death blocked | n |
|---|---|---|
| +/+ | 0 | 6 |
| psf-2(t3443ts) | 67 | 6 |
| psf-2(t3443ts);psf-2(+) | 0 | 6 |

**D**

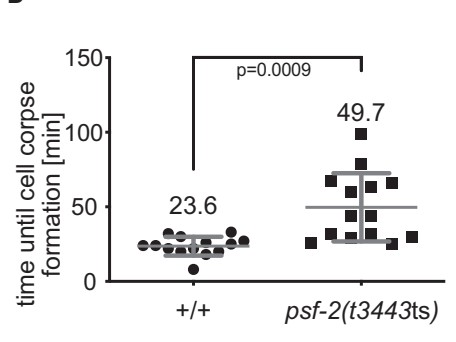

**E**

| Genotype | % QL.pp cell death blocked | n |
|---|---|---|
| +/+ | 0 | 60 |
| psf-2(t3443ts)15°C | 0 | 77 |
| psf-2(t3443ts) | 29 | 75 |

with cell cycle lengths close to the average cell cycle length of wild-type mothers (42 min) and there are mothers with cell cycle lengths almost four times the average cell cycle length of wild-type mothers (168 min). The independence of the daughter cell fate from the cell cycle length of the mother cell is furthermore exemplified in Fig. 3C, which depicts 4D lineaging data of two of the 1st wave cell deaths, ABalaapapa (referred to as "CD1") and ABalaappaa (referred to as "CD2"). In the wild-type embryo (+/+), both cells died within 20–30 min after the completion of their mothers' divisions, and the cell cycle length of their mothers was 42 min and 44 min, respectively. In the *psf-2(t3443ts)* embryo, the cell cycle length of the two mother cells was increased to 97 min (mother of CD1) and 107 min (mother of CD2), respectively. CD1 failed to die (indicated in red) whereas CD2 died 20–30 min after the completion of its mother's division. (Of note, the

**Fig. 2 | Reducing *psf-2* GINS2 function causes a general block in cell death. A** Cell fate analysis of the first 13 cell deaths of the AB lineage in wild-type (+/+) (*n* = 4; data from one representative embryo is shown), *psf-2(t3443*ts) (*n* = 4), *psf-2(t3443*ts); *psf-2*(+) (transgene *bcEx1302*) (*n* = 4) and *psf-2(RNAi)* (*n* = 3). Cell fate was determined based on 4D lineaging analyses performed on long-term recordings done at 25 °C as described in "Methods". In the case of *psf-2(t3443*ts); *psf-2*(+), data shown is from embryos that hatched. **B** Percentage [%] cell death blocked during first wave of cell death (13 AB-derived cell deaths). Summary of data shown in (**A**) and data for the following additional genotypes: *ced-3(n717)*, *psf-3(RNAi)*, *mcm-7(RNAi)*, *mcm-2(RNAi)* and *tyms-1(e2300*ts). Unless noted otherwise, recordings were performed at 25 °C. **C** Percentage [%] MSpaapp cell death blocked in wild-type (+), *psf-2(t3443*ts), and *psf-2(t3443*ts); *psf-2*(+) (transgene *bcEx1302*) animals. In the case of *psf-2(t3443*ts); *psf-2*(+), data shown is from embryos that hatched. **D** Time until cell corpse formation [min] in wild-type (+/+) and *psf-2(t3443*ts) animals. Time measured was from the birth of the cell until the cell formed a cell corpse (button-like appearance by DIC) (*n* = 14 for both genotypes). Mean ± SD are indicated, and mean value is stated above data. *P* value = 0.0009, unpaired two-tailed t-test with Welch correction. **E** Percentage [%] QL.pp cell death blocked in wild-type (+/+) animals at 25 °C and *psf-2(t3443*ts) animals at 15 °C and 25 °C using the *bcIs133* (P<sub>toe-2</sub>*gfp*) transgene.

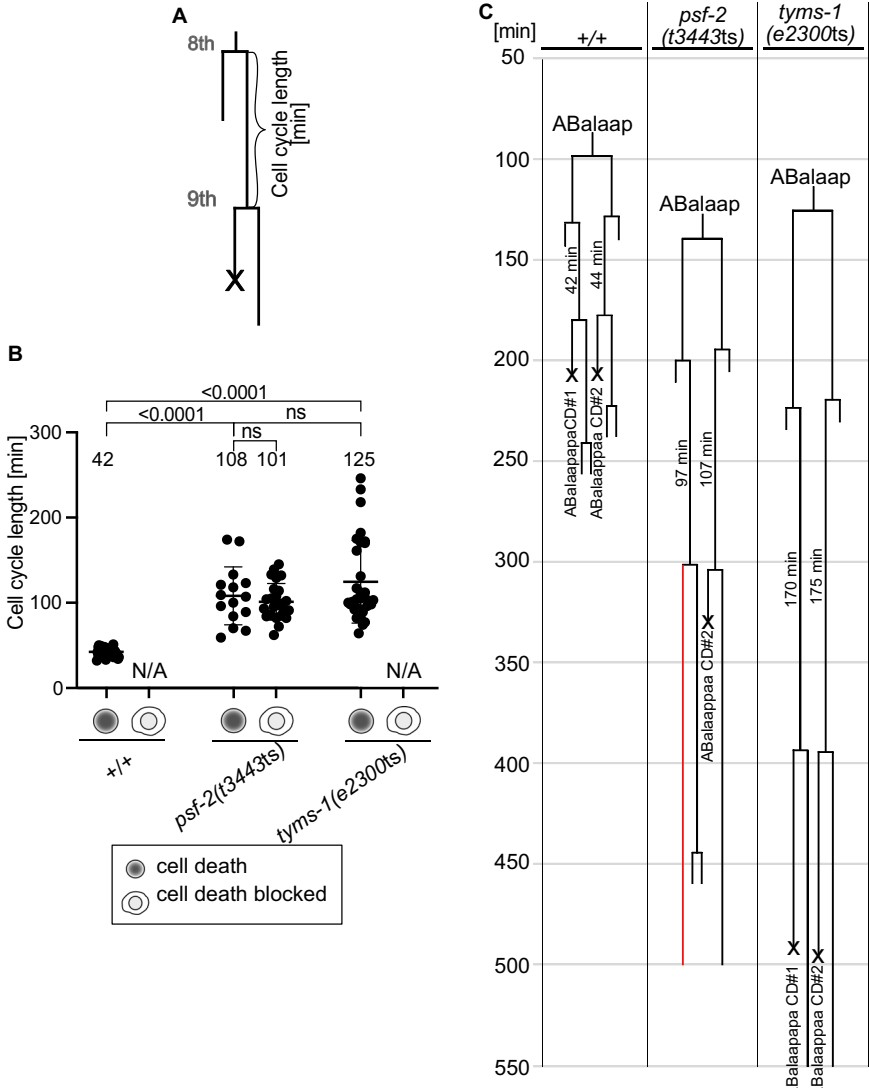

**Fig. 3 | Lack of correlation between the increased cell cycle length phenotype and the cell death phenotype of *psf-2(t3443*ts*)* animals. A** Schematic of cell cycle length [min] measurements of mothers of cells that die. "Cell cycle length" is defined as the time in minutes from their births after the 8th round of cell division until their own divisions (9th round of cell division). Data were generated from 4D lineaging analyses done on long-term recordings performed at 25 °C. **B** Cell cycle length [min] of mothers of cells that die in wild type (+/+) (*n* = 56), *psf-2(t3443*ts) (mothers whose daughters died ("cell death") (*n* = 15)) and mothers whose daughters failed to die ("cell death blocked") (*n* = 25)) and *tyms-1(e2300*ts) (*n* = 31). The average cell cycle lengths are stated above the data. Mean ± SD are indicated, and value of mean is stated above data. *P* value **** <0.0001, unpaired two-tailed t-test with Welch correction was used; ns: not significant. N/A not applicable as in this genetic background no cell death was blocked. **C** ABalaap lineage of representative animals of the genotypes indicated. The cell death 1 and 2 (CD#1 and #2) are indicated. A red lineage indicates a block in division.

sister of CD1 divided like in wild type, but the sister of CD2, failed to divide (Fig. 3C).

In summary, in *psf-2(t3443*ts*)* animals, there is no correlation between the cell cycle length of a mother cell and the ability of its daughter to die. This suggests that the roles of *psf-2* in DNA replication (the likely cause of the increased cell cycle length) and in the acquisition of the cell death fate represent two independent activities of *psf-2* GINS2. Our finding that the Emb phenotype and the Ced phenotype observed in *psf-2(t3443*ts*)* animals differ in penetrance and expressivity (see Figs. 1 and 2) furthermore supports this notion.

## Role of *tyms-1* TYMS in embryonic development and cell death

The lack of correlation between the increased cell cycle length phenotype and the Ced phenotype observed in *psf-2(t3443*ts*)* mutants suggests that the role of PSF-2 GINS2 in the acquisition of the cell death fate is separable from its role in DNA replication and thus presumably its DNA helicase activity. To explore this notion further we sought to abrogate DNA replication and increase cell cycle length more directly. Like *psf-2(t3443*ts*)* embryos, at the non-permissive temperature (25 °C), embryos homozygous for the temperature-sensitive mutation *e2300*ts initiate morphogenesis but then arrest[35] (Suppl Fig. 5A). At 25 °C, *e2300*ts embryos exhibit a fully penetrant Emb phenotype and a reduced brood size; however, at the permissive temperature (15 °C), *e2300*ts animals are essentially indistinguishable from wild type (Suppl Fig. 5C, D). We found that *e2300*ts is a partial loss-of-function mutation of the gene *tyms-1*, which encodes an ortholog of human thymidylate synthase TYMS[36] (see Methods). Specifically, *e2300*ts is a missense mutation that causes a guanine to thymine change at position 240 (G240T) of the coding region of the *tyms-1* gene. This results in a

predicted tryptophan to cysteine change at position 80 of the amino acid sequence of the TYMS-1 protein (W80C), which is a residue that is conserved between *C. elegans*, mouse and human (Suppl Fig. 5E). Thymidylate synthase is the sole enzyme capable of de novo synthesis of thymidine nucleotide precursors, and its inactivation for instance through inhibitors such as 5-fluorouracil (5-FU) leads to a block in DNA replication and a block in cell division[37,38]. As shown in Figs. 1C, D, 4D lineaging analyses revealed that *tyms-1(e2300*ts*)* embryos exhibit increased cell cycle lengths in the five consecutive rounds of cell divisions starting from ABarp that give rise to ABarpppppp, very similar to what we observed for *psf-2(t3443*ts*)* embryos. Likewise, *tyms-1(e2300*ts*)* animals exhibit increased cell cycle lengths in the six consecutive rounds of cell divisions starting from MS that give rise to MSpppppp (Suppl. Fig. 3A, B). Next, we analyzed 1st wave cell deaths derived from AB in three *tyms-1(e2300*ts*)* animals and found that in contrast to *psf-2(t3443*ts*)* animals, 0% of the cell deaths are blocked (Fig. 2B, *tyms-1(e2300*ts*)). We also determined the cell cycle lengths of the mother cells of these 1st wave cell deaths and found that the

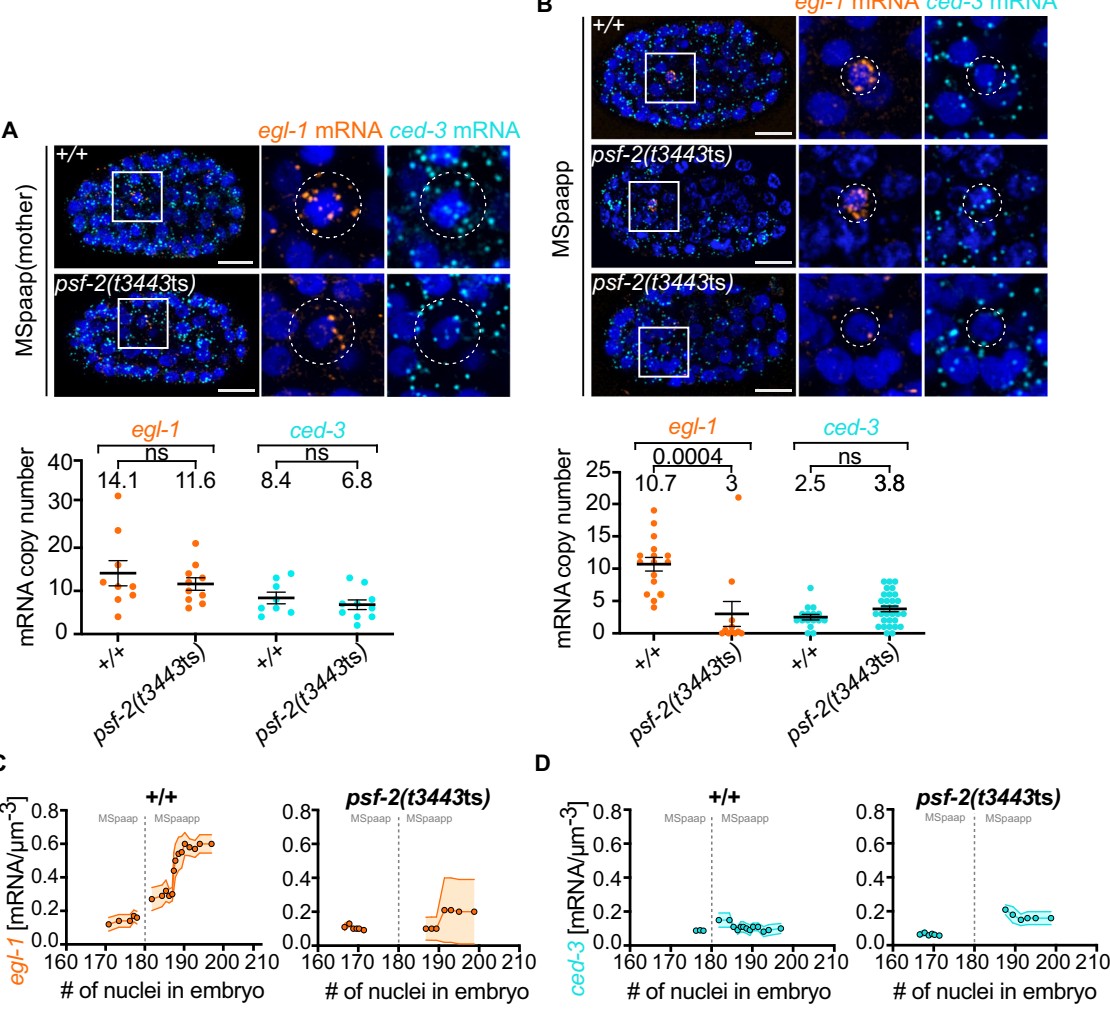

**Fig. 4 | Reducing *psf-2* GINS2 function abolishes the increase in *egl-1* BH3-only mRNA observed in daughters that die.** smRNA FISH analysis in (**A**) MSpaap (mother) and (**B**) MSpaapp (daughter that dies) in wild-type (+/+) and *psf-2(t3443*ts*)* embryos. *Top* Representative fluorescent images of embryos and MSpaapp or MSpaapp (insets and enlargements). Cells are indicated by white circles and are 6.0 μm (MSpaap) and 3.5 μm (MSpaapp) in diameter. Nuclei are labeled with DAPI and are shown in dark blue. Labeled *egl-1* mRNAs or *ced-3* mRNAs are shown in orange or light blue, respectively. Scale bars 10 μm. *Bottom*. mRNA copy numbers in MSpaap (*egl-1*: +/+ n = 9; *psf-2(t3443*ts*)* n = 10; *ced-3*: +/+ n = 8; *psf-2(t3443*ts*)* n = 10) and

MSpaapp (*egl-1*: +/+ n = 16; *psf-2(t3443*ts*)* n = 11; *ced-3*: +/+ n = 16; *psf-2(t3443*ts*)* n = 31). Mean ± SEM are indicated, and mean value is shown above data. *P* value = 0.004, ns: not significant. A two-tailed Mann-Whitney test was performed. Time course of mRNA concentration [copy number/μm³] of (**C**) *egl-1* mRNA and (**D**) *ced-3* mRNA in MSpaap and MSpaapp in wild-type (+/+) and *psf-2(t3443*ts*)* embryos. X-axis indicates number of nuclei in embryo. As indicated by the vertical dotted line, MSpaap divides when ~180 embryonic nuclei are present in the embryo. Graphs were generated from raw data as a centered moving average of order 5 as described in "Methods". Shaded areas represent SEM.

average cell cycle length was increased to 125 min, which is more than three times longer than the average cell cycle length of these mothers in wild type (Fig. 3B). To give an example, Fig. 3C depicts 4D lineaging data for CD1 and CD2 in one *tyms-1(e2300*ts) embryo. The cell cycle length of the CD1 and CD2 mother cells in this embryo was 170 min and 175 min, respectively, but CD1 and CD2 both died.

In summary, in line with the importance of thymidylate synthase for DNA synthesis and, hence, DNA replication, reducing *tyms-1* TYMS function causes increased cell cycle lengths, ultimately resulting in embryonic lethality. However, in contrast to the reduction of *psf-2* GINS2 function, reducing *tyms-1* TYMS function does not cause a block in the cell death fate. Together with the results presented in Fig. 3C, these results demonstrate that increasing cell cycle lengths in mother cells (presumably through either compromising DNA unwinding or DNA synthesis) is not sufficient to block the deaths of daughters that die. Therefore, the role of *psf-2* GINS2 in the acquisition of the cell death fate is separable from its role in DNA replication.

### Role of *psf-2* GINS in *egl-1* transcriptional upregulation

Most of the cell deaths that occur during *C. elegans* development (including all cells that die during the 1st wave of cell death) are apoptotic cell deaths and dependent on the gene *egl-1*, which encodes a BH3-only protein[18,19]. In contrast to the genes that act downstream of *egl-1* in the apoptosis pathway (i.e., *ced-9* Bcl-2, *ced-4* Apaf-1, *ced-3* caspase) and that are ubiquitously expressed at least during embryogenesis[39–41], *egl-1* expression is mainly restricted to cell death lineages[18,23]. Using smRNA FISH, we previously showed that within a cell death lineage, a low concentration of *egl-1* mRNA is detected in the "mother" cell. The mother cell divides asymmetrically by both fate and size. The resulting smaller daughter cell dies, whereas the larger one survives. (Daughter cell size ratios differ between cell death lineages and range from about 5.0 to 1.5.) Immediately after mother cell division, *egl-1* mRNA concentrations in the two daughters are similar to *egl-1* mRNA concentrations in the mother cell. Within a few minutes, however, *egl-1* mRNA concentration decreases to undetectable levels in the daughter that survives but increases several-fold in the daughter that dies[23]. Based on *egl-1* transcriptional reporters, the increase in *egl-1* mRNA concentration observed by smRNA FISH in the daughter that dies is most probably the result of an increase in *egl-1* transcription[21,25].

To determine the impact of reducing *psf-2* GINS2 function on *egl-1* expression, we analyzed *egl-1* mRNA levels in the MSpaap lineage using smRNA FISH. The MSpaap mother cell divides to give rise to MSpaapa, which survives, and MSpaapp, which dies. As described above, we found that in *psf-2(t3443*ts) animals, 67% of MSpaapp deaths are blocked (Fig. 2C). In wild-type animals, we found on average 14.1 and 10.7 *egl-1* mRNAs in MSpaap and MSpaapp, respectively (Fig. 4A, B, +/+, *egl-1*). (Of note, the estimated cell volumes of MSpaap and MSpaapp are 113 μm³ and 22 μm³, respectively[23]. MSpaap is therefore about 5-times the volume of MSpaapp.) To assess the temporal dynamics of *egl-1* mRNA concentration in MSpaap and MSpaapp, we determined the developmental stage of the embryos analyzed based on the number of nuclei in the embryo (see Methods). As seen in Fig. 4C (+/+), we found that in MSpaap, *egl-1* mRNA concentration between embryonic nuclei stages 170 and 177 is 0.1 to 0.2 mRNAs/μm³. In MSpaapp, *egl-1* mRNA concentration increases from about 0.2 mRNAs/μm³ at embryonic nuclei stage 182 to about 0.6 mRNAs/μm³ at embryonic nuclei stage 197. In *psf-2(t3443*ts) animals, we found on average 11.6 *egl-1* mRNAs in MSpaap, which is similar to what we detected in MSpaap in wild type (Fig. 4A, *psf-2(t3443*ts), *egl-1* mRNA). However, in contrast to wild type, we only found on average 3.0 *egl-1* mRNAs in MSpaapp. Indeed, in eight out of 11 embryos (about 73%), essentially no *egl-1* mRNA was detectable in MSpaapp (Fig. 4B, *psf-2(t3443*ts), *egl-1*). (In the case of these eight embryos, we measured *egl-1* mRNA in four cells at the position where MSpaapp is usually located, determined the average *egl-1* mRNA copy number of these four cells and used this as a value

for MSpaapp.) Importantly, in these eight embryos, we did not detect any cells in the position expected for MSpaapp that showed increased levels of *egl-1* mRNAs. This supports the notion that the absence of *egl-1* mRNAs in MSpaapp in *psf-2(t3443*ts) animals is the result of the loss of *egl-1* expression in MSpaapp rather than a switch in daughter cell fates and the concomitant "gain of *egl-1* expression" in MSpaapp's sister cell MSpaapa (see Fig. 4B). In addition, we found that compared to wild type, *egl-1* mRNA concentration and dynamics in MSpaap are largely unchanged in *psf-2(t3443*ts) animals (0.1 mRNAs/μm³) (Fig. 4C, *psf-2(t3443*ts)); however, the burst of *egl-1* mRNA concentration observed in MSpaapp between embryonic nuclei stages 182 and 197 is essentially absent in *psf-2(t3443*ts) animals. Of note, we did not detect changes indicative of ectopic expression in the pattern of *egl-1* mRNA in *psf-2(t3443*ts) embryos, suggesting that the loss of *psf-2* does not cause general mis-regulation of *egl-1* expression (see Fig. 4A, B, *psf-2(t3443*ts), *egl-1*). As a control, and to determine whether reducing *psf-2* function affects the expression of genes other than *egl-1*, we analyzed *ced-3* mRNA numbers and concentrations in the same cells through double-labeling. As seen in Fig. 4A, B (*ced-3* mRNA), there is no significant difference between wild-type and *psf-2(t3443*ts) animals in the average *ced-3* mRNA numbers in MSpaap (8.4 and 6.8) or MSpaapp (2.5 and 3.8). In addition, between embryonic nuclei stages 170 to 200, *ced-3* mRNA concentrations in MSpaap and MSpaapp are relatively stable in both wild-type and *psf-2(t3443*ts) animals (Fig. 4D).

In summary, reducing *psf-2* GINS2 function abolishes the increase in *egl-1* BH3-only mRNA observed immediately after MSpaap division in MSpaapp, but does not affect *ced-3* caspase mRNA levels in either MSpaap or MSpaapp. These results suggest that in cell death lineages, *psf-2* is specifically required for the increase in *egl-1* mRNA levels in daughters that die. Thus, our results suggest that *psf-2* GINS2 is required for cell fate divergence in cell death lineages.

### Role of *psf-2* GINS2 in other cell fates

The cell ABaraappaa ("MI mother cell") divides during the 10th round of cell division and generates an anterior daughter, ABaraappaaa, which differentiates into the pharyngeal motor neuron/interneuron MI, and a posterior daughter cell, ABaraappaap, which differentiates into the pharyngeal marginal cell m1DR ("MI/m1DR decision")[9]. A cold-sensitive (cs) gain-of-function (gf) mutation of the gene *his-9*, *n5357*gf, one of 14 genes in the *C. elegans* genome that encode replication-dependent histone H3, has been proposed to interfere with H3-H3 interactions and, hence, the formation of H3-H4 tetramers and nucleosome assembly[42]. Importantly, it was previously shown that in *his-9(n5357*gf) animals, instead of differentiating into the MI neuron, ABaraappaaa adopts the fate of the pharyngeal epithelial cell e3D[42]. Based on this, Horvitz, Stillman and co-workers proposed that the ability of ABaraappaa to adopt the MI fate is dependent on epigenetically distinct sister chromatids that are generated in the MI mother cell through unequal nucleosome transfer during replication and that are selectively segregated into the two daughter cells[42]. To determine whether the role of *psf-2* GINS2 in cell fate divergence extends to cell fate decisions that, first, do not generate a daughter that dies and, second, have been proposed to involve epigenetically distinct chromatids generated through replication-coupled nucleosome assembly, we, next, analyzed the MI/m1DR decision. Using a MI-specific reporter (P$_{sams-5}$gfp)[43], we found that 100% of wild-type animals have one MI neuron, but only 51% of *his-9(n5357*gf) animals raised at non-permissive temperature (15 °C) have an MI neuron[42] (Fig. 5A, *his-9(n5357*gf)). To determine whether reducing *psf-2* function affects the MI fate, we analyzed *psf-2(t3443*ts) animals at the non-permissive temperature. We found that 89% of *psf-2(t3443*ts) animals have one MI neuron and 5% have no MI neuron (Fig. 5A, B, *psf-2(t3443*ts)). Interestingly, based on reporter expression, 6% of the animals had an additional MI neuron ("ectopic MI"). Therefore, reducing *psf-2* GINS2 function can impact the MI/m1DR decision−albeit at low penetrance−and,

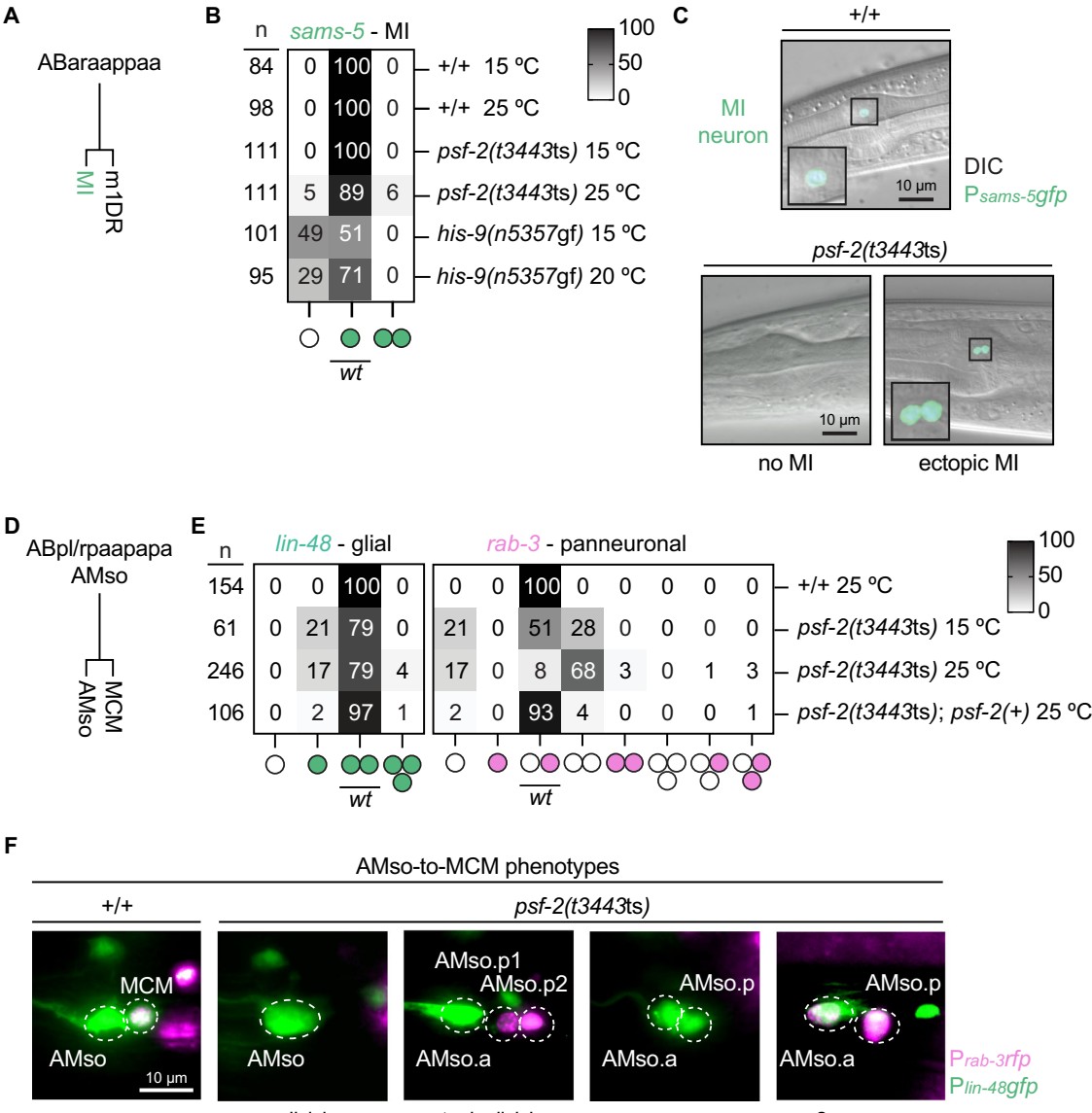

**Fig. 5 | Reducing *psf-2* GINS2 induces cell fate defects in the MI and AMso lineages. A** Embryonic cell lineage of the MI neuron. **B** Heat map showing the percentage of L4 hermaphrodites that express the MI-specific reporter transgene P*sams-5gfp* (*nls396*) in *psf-2(t3443*ts) and *his-9(n5357*gf) mutants. Rows represent different genotypes and temperatures. Columns represent different phenotypes depicted as circles (white circles represent cells without expression, green circles represent cells expressing the P*sams-5gfp* reporter). *n* indicates the number of animals scored. For each genotype, the experiment was repeated at least twice independently (technical replicates). **C** Differential interference contrast (DIC) and fluorescence micrographs of the anterior pharynx of hermaphrodites showing expression of the MI-specific transgene P*sams-5gfp* in wild type and *psf-2(t3443*ts) mutants grown at the non-permissive temperature of 25 °C. These micrographs represent examples of the phenotypes described in (**B**). **D** Postembryonic cell lineage of the AMso glial cell and the MCM neuron. **E** Heat map showing the

percentage of cells per side in adult males that express the glial reporter transgene P*lin-48gfp* (*sals14*; left panel) and the panneuronal marker P*rab-3NLS::rfp* (*otls356*; right panel) in various genotypes and temperatures (white circles represent cells without expression, green circles represent cells expressing the P*lin-48gfp* reporter and red circles represent cells expressing the P*rab-3NLS::rfp* reporter). *bcEx1306* was used for *psf-2(t3443*ts) rescue. In wild-type animals, P*lin-48gfp* expression is transiently observed in the posterior cell (MCM) and absent in adults; this allows the division to be monitored[98]. *n* (number of cells per side) corresponds to the average of two independent experiments (technical replicates) for every genotype, with similar results. **F** Fluorescence micrographs showing expression of P*lin-48gfp* and P*rab-3NLS::rfp* in the AMso and MCM cells of wild type and *psf-2(t3443*ts) adult animals grown at the non-permissive temperature of 25 °C. These micrographs represent examples of the phenotypes described in (**E**).

hence, a process that has been proposed to involve epigenetically distinct chromatids generated through unequal replication-coupled nucleosome assembly.

The AMso cells (ABpl/rpaapapa) are a pair of bilaterally symmetrical glial cells that are born in the embryo. In males, each of the AMso cells divides asymmetrically during post-embryonic development to generate another glial cell (ABpl/rpaapapaa referred to as "AMso") and a MCM neuron (ABpl/rpaapapap) ("AMso/MCM decision")[44]. Next, we

decided to analyze the AMso/MCM decision to determine whether the role of *psf-2* GINS2 in cell fate divergence, first, extends to cell fate decisions that occur during post-embryonic development and, second, is required in mother cells. In addition, the availability of compatible reporters allowed us to simultaneously monitor both daughter cell fates. We analyzed *psf-2(t3443*ts) animals that had been shifted to the non-permissive temperature (25 °C) during the first larval stage of development (i.e., after the mother cells ABpl/rpaapapa had been

generated) using the glia-subtype reporter P*lin-48*g*fp*[45] and the pan-neuronal reporter P*rab-3*NLS::*rfp*[46]. As expected, given the role of *psf-2* in DNA replication, 17% of the AMso divisions are blocked in *psf-2(t3443*ts*)* animals shifted to the non-permissive temperature (Fig. 5E and F). Importantly, we found that in the cases where the AMso did divide, there was no *rab-3* expression in 82% of the cases (167/204 AMso divisions). This lack of neuronal differentiation in the posterior daughter cell accounts for 68% of the total population of AMso divisions analyzed (Fig. 5E right panel, *psf-2(t3443*ts*)* 25 °C). The *rab-3*-lacking cells retained *lin-48* expression into adulthood (Fig. 5E and F "no neuron"). In 3% of the cases, we also observed *rab-3* expression in both the anterior and posterior daughter cells (Fig. 5D "2 neurons"). Importantly, we never observed a switch of the glial and neuronal fates, i.e., cases in which the anterior daughter adopted the neuronal fate (expressing the P*rab-3*NLS::*rfp* reporter) and the posterior daughter adopted the glial fate (expressing only the P*lin-48*g*fp* reporter). This supports the notion that *psf-2(t3443*ts*)* causes the loss of the neuronal fate, rather than a switching of daughter cell fates, within the AMso lineage. Interestingly, at the non-permissive temperature, 4% of the cells divide ectopically to produce an extra cell (7/8 ectopic cells express the neuronal marker *rab-3*). Finally, all defects observed were rescued by *psf-2*(+). Therefore, *psf-2* GINS2 is required for the acquisition of the MCM fate during the AMso/MCM decision in males, which takes place post-embryonically, and most likely acts in the ABpl/rpaapapa mothers rather than other progenitors.

In summary, based on our analyses of the MI/m1DR and AMso/MCM decision, we conclude that the role of *psf-2* GINS2 in cell fate divergence is not restricted to cell fate decisions that generate a cell death or that take place during embryonic development.

## Discussion

Genetic studies of programmed cell death in *C. elegans* have been instrumental in the elucidation of the conserved apoptosis pathway[18,19]. Continuing in this tradition, we have discovered that CMG (Cdc45-MCM2-7-GINS) is required for cell death during *C. elegans* development. CMG is unlike others factor that have been identified as required for cell death during *C. elegans* development, because it is essential for embryonic viability. Importantly, we demonstrate that CMG's role in the acquisition of cell fate is not specific to the cell death fate. As outlined below, our genetic studies of programmed cell death have uncovered a CMG-mediated mechanism that is fundamentally important for *C. elegans* embryonic and post-embryonic development and that is likely conserved in higher organisms.

### CMG has a conserved role in the divergence of cell fates in the context of asymmetric cell division

How daughter cells acquire different fates during asymmetric cell division is incompletely understood[1–4]. Here we demonstrate that CMG, a core complex of the eukaryotic replisome, is required for the asymmetric acquisition of at least three different cell fates in *C. elegans* (cell death fate, MI fate, MCM fate), suggesting a widespread role for CMG in this process. The groups of Zhiguo Zhang, Anja Groth, and Haiyun Gan have recently independently reported that Mcm2, a component of CMG, or POLE-3/POLE-4, a subcomplex of the leading strand polymerase Polε, are required for the ability of mouse embryonic stem cells (ESCs) to transition from a naïve cell state to a differentiated cell state[47–50]. Importantly, our results not only confirm their in vitro data but provide in vivo evidence for a role of CMG in the divergence of cell fates in the context of asymmetric cell division. Whether the depletion of POLE-3/POLE-4 also affects cell fate divergence in *C. elegans* remains to be determined. Our results also provide support for the quantal cell cycle theory, which proposes that genome alterations that occur during cell division enable the expression of distinct sets of genes in the daughter cells[5,6].

Although coupled to replication, the function of Mcm2 in the differentiation of mouse ESCs in vitro is separable from its helicase activity and hence, independent of its well described role in DNA replication[48–50]. Similarly, we present in vivo evidence in support of the notion that the role of *C. elegans* CMG in cell fate divergence is separable from its role in DNA replication and thus presumably DNA helicase activity. Specifically, (1) increases in cell cycle length in *psf-2(t3443*ts*)* mutants (likely reflecting replication stress) do not correlate with the loss of the cell death fate and (2) increases in cell cycle length in *tyms-1(e2300*ts*)* mutants (also likely reflecting replication stress) do not lead to the loss of the cell death fate. Together, these findings suggest that in the context of asymmetric cell division, CMG has a conserved role in the divergence of cell fates that is distinct from its conserved role in DNA replication. Finally, the idea that CMG has a conserved role in cell fate is supported by previous observations implicating Psf2 GINS2 in nervous system development in Xenopus and zebrafish[51,52], Sld5 GINS4 in early embryonic development and nervous system development in mouse[53,54] and MCM5 in neuronal differentiation in Drosophila[55].

Interestingly, genes encoding components of the pre-replication complex, such as *orc-1-5* or *mcm-2-7*, but not genes encoding components required for the assembly of the functional replicative helicase, such as *psf-1*, *psf-2*, *psf-3*, or *sld-5*, are required for the expression of pro-invasive genes during anchor cell invasion in *C. elegans* larvae[56]. This provides additional evidence that the replisome is involved in regulating gene expression. However, it is important to note that the anchor cell is a non-dividing post-mitotic cell. The role of the pre-replication complex in the expression of pro-invasive genes is therefore independent of DNA replication and, hence, independent of cell division.

### Working model for how *C. elegans* CMG causes divergence of the cell death fate

Apart from their known roles as part of the replicative helicase or the leading strand polymerase Polε, Mcm2 and POLE3/POLE4 bind histones and act as histone chaperones during replication[57–63]. In this capacity, they transfer parental nucleosomes carrying epigenetic marks on their histone H3 and H4 moieties from the parental chromosome to the lagging or leading strand, respectively[63,64]. The roughly equal inheritance of parental nucleosomes to leading and lagging strand is thought to be critical for the maintenance of gene expression patterns, and thus cell fate stability, during symmetric or "proliferative" cell divisions[65–67]. Indeed, the loss of Mcm2's histone chaperone activity or the depletion of POLE-3 or POLE-4 in mouse ESCs in vitro has recently been reported by the Groth, Gan and Zhang groups to disrupt the equal inheritance of parental nucleosomes to leading and lagging strands and to result in sister chromatids with alternating, complementary patches of parental or new nucleosomes along the chromosomes[48–50]. Unexpectedly, the ability of these mutant ESCs to undergo self-renewing proliferative cell divisions was essentially unaffected; however, as mentioned above, defects were observed in the ability of these cells to acquire a differentiated state. Bivalent genes are considered key regulators during differentiation. They are decorated by nucleosomes that carry both the repressive mark H3K27me3 and the activating mark H3K4me3 and this "bivalent" condition is thought to enable rapid transcriptional activation in a cell- or lineage-specific manner[68,69]. Based on transcriptome profiling combined with genome-wide analyses of epigenetic marks, it was proposed that the defects in differentiation observed in the Mcm2 and POLE-3/POLE-4 mutant ESCs are in part the result of the over-enrichment of the repressive epigenetic mark H3K27me3 at bivalent genes, which impairs activation of these genes during differentiation[48–50].

We demonstrate that reducing *C. elegans* CMG function results in the inability to increase transcription of the cell death fate determinant

*egl-1* in daughters that die immediately following mother cell division. This establishes the BH3-only gene *egl-1* as a target of CMG. Based on the following observations, we propose that CMG impacts *egl-1* BH3-only expression at the epigenetic level. First, our results suggest that the cell death defect observed in response to reducing CMG is independent of CMG's replication-coupled helicase activity. We acknowledge that it does remain formally possible that CMG helicase activity is required to remove torsional tension during transcription and that this hypothetical role could account for the effects of CMG on gene expression. However, we consider it most likely that the role of CMG in cell death is dependent on CMG's recently described histone chaperone activity. Second, the mutation *his-9(n5357*gf), which interferes with the formation of H3·H4 tetramers, results in the loss of the MI fate[42] (see Fig. 5). We show that reducing CMG also results in the loss of the MI fate albeit at low penetrance. Third, the *egl-1* BH3-only locus on LGV spans ~15 kb and includes the *egl-1* transcription unit as well as extensive *cis*-acting elements (i.e., enhancers), some of which are located beyond transcription units upstream and downstream of the *egl-1* transcription unit[18,21] (Suppl. Fig. 6). We have mined publicly available data on genome-wide distributions of nucleosomal histone H3 post-translational modifications (PTMs) in *C. elegans* embryos and found that the *egl-1* locus is decorated by both nucleosomes carrying repressive H3K27me3 marks and nucleosomes carrying activating H3K4me3 marks (bulk embryo data representative of non-cell death lineages)[70,71] (Suppl. Fig. 6). It remains to be ascertained whether H3K27me3 and H3K4me3 are present on the same nucleosomes within a single cell; however, considering that *egl-1* BH3-only is a key developmental regulator that is expressed in a highly dynamic and spatiotemporally restricted manner, this makes it likely that *egl-1* BH3-only represents a bivalent gene whose expression during differentiation is affected when the equal inheritance of parental nucleosomes is disrupted during replication. (In addition, in bulk embryo data representative of non-cell death lineages, the *egl-1* locus is devoid of nucleosomes carrying the activating H3K36me3 mark but is decorated with nucleosomes carrying the mark H3K4me1[70]. The *egl-1* locus can therefore be considered "poised" for transcriptional activation (Suppl. Fig. 6)). Furthermore, based on the following observations, we propose that CMG acts during mother cell replication to enable epigenetic changes at the *egl-1* BH3-only locus. First, Nakano et al. demonstrated that the mutation *his-9(n5357*gf) acts in the MI mother cell to cause loss of the MI fate[42]. Second, we demonstrate that a shift - in *psf-2(t3443*ts) animals - of the AMso glial cell to the non-permissive temperature (25 °C) is sufficient to cause AMso to divide symmetrically and generate two AMso-like cells rather than divide asymmetrically and generate one AMso glial cell and one MCM neuron (Fig. 5).

Based on these observations, we propose that CMG enables epigenetic changes at the *egl-1* BH3-only locus during mother cell replication. These changes are mediated by CMG's histone chaperone activity and, hence, its role in replication-coupled nucleosome assembly, and they are required for the increase in *egl-1* expression in daughters that die. The nature of the relevant epigenetic changes remains to be determined but may include the removal of the repressive mark H3K27me3. According to this working model, reducing CMG function would prevent these epigenetic changes and thus eliminate the increase in *egl-1* expression in daughters that die. The development of methods for in vivo visualization of the chromatin state at the *egl-1* BH3-only locus in specific cell death lineages in the developing embryo will be necessary to test our model[72], and this is one of our prime goals in the future.

### How is symmetry of *egl-1* BH3-only expression broken during asymmetric cell divisions?

The working model described above explains how *egl-1* BH3-only transcription can be activated in daughter cells. What remains unclear is how asymmetry of *egl-1* transcriptional activation is created to cause *egl-1* expression in one, but not the other daughter cell. As mentioned above, the non-random segregation of direct repressors of *egl-1* transcription into the daughter that survives may contribute to the repression of *egl-1* expression in these cells[24,25]. Therefore, one possible model is that CMG-dependent epigenetic changes that enable transcriptional activation at the *egl-1* locus occur on both sister chromatids. In this scenario, both daughter cells would inherit two *egl-1* loci that are competent for transcriptional activation, but transcriptional activation would be blocked in the daughter that survives, because of the presence of a direct repressor of *egl-1* transcription. In this scenario, symmetry breakage would be caused by the non-random segregation of *trans*-acting factors, e.g., transcriptional repressors. Alternatively, symmetry breakage could occur at the level of the *egl-1* locus itself. Specifically, the CMG-dependent epigenetic changes at the *egl-1* locus could occur on only one of the two emerging sister chromatids and the two *egl-1* loci competent for transcriptional activation could be inherited specifically by the daughter that dies. How CMG-dependent epigenetic changes at the *egl-1* locus could occur on only one of the two emerging sister chromatids is currently unknown. Interestingly, there is some evidence that epigenetically distinct sister chromatids can be generated and non-randomly segregated during asymmetric or "informative" cell divisions in stem cell lineages. Specifically, the group of Xin Chen has reported that in asymmetrically dividing *D. melanogaster* germline and intestinal stem cells and mouse ESCs, distinct sister chromatids with either parental nucleosomes or new nucleosomes are non-randomly segregated into the self-renewing stem cell daughter or differentiating daughter, respectively[73–76]. Furthermore, Zhiguo Zhang's group has recently shown that parental nucleosomes carrying the repressive mark H3K9me3 are preferentially segregated to the leading strand during the replication of LINE1 transposable elements and that this non-random segregation of nucleosomes is dependent on the interaction of the Human Silencing Hub (HUSH) complex with POLE-3/POLE-4[77]. Additionally, scenarios can be envisioned in which both *trans*-acting factors and epigenetically distinct *egl-1* loci could be non-randomly segregated. Finally, regardless of the specific scenario, they all require that the mother cell becomes polarized, which is expected to be dependent on extrinsic factors such as intercellular signals.

### Control of gene expression in multilineage priming

In multilineage priming, genes that determine the fates of the two daughter cells are co-expressed in the mother cell, but after mother cell division, their expression becomes restricted to one or the other daughter cell[10]. Our results uncover an intrinsic mechanism through which the expression of a *C. elegans* cell fate determinant that is subject to multilineage priming is maintained and increased in one of the daughter cells. As outlined above, we propose that CMG-dependent epigenetic changes at the *egl-1* BH3-only locus during replication in the mother cell enable increased expression of *egl-1* in the daughter that dies. Whether CMG is required for changes in the expression of other *C. elegans* genes that exhibit multilineage priming remains to be determined[10]. It also remains to be determined whether CMG is required for the expression of genes that are subject to multilineage priming in other organisms[11–14,16]. Our working model proposes that CMG controls *egl-1* expression at the epigenetic level in most if not all cell death lineages i.e., globally. *egl-1* expression has previously been shown to be controlled at both the transcriptional and post-transcriptional level. Briefly, lineage-specific transcription factors act through specific *cis*-acting elements in the *egl-1* locus to control *egl-1* transcription in specific cell death lineages (see Suppl. Fig. 6) and the loss of a lineage-specific transcription factor or the loss of the *cis*-acting element through which it acts results in the block of one or a few specific cell deaths (in the case of transcriptional activators) or the ectopic death of specific cells that normally survive (in the case of transcriptional repressors)[18,78–80]. At the post-transcriptional level,

members of the miR-35 and miR-58 families of microRNAs act through the 3′UTR of *egl-1* mRNAs to globally ensure that the level of *egl-1* expression in mothers of cells that die does not reach the level necessary to trigger cell death, and the loss of miR-35 and miR-58 microRNAs causes mother cells to die precociously[23,81]. How control of *egl-1* expression at the epigenetic, transcriptional and post-transcriptional levels is coordinated within a particular cell death lineage to reproducibly results in the highly dynamic and spatio-temporally restricted pattern of expression observed remains to be investigated. It also remains to be investigated whether such complex control of gene expression at multiple levels underpins the expression of other *C. elegans* genes that are subject to multilineage priming.

Multilineage priming is a phenomenon also observed during mammalian development and is well characterized during lineage commitment in the hematopoietic system[13,14,17]. Inborn errors of immunity (IEI) are rare genetic conditions that are characterized by the absence or dysfunction of specific types of immune cells[82], such as for example Natural Killer (NK) cells[83–85]. Importantly, of the six genes that have so far been identified to mutate and cause NK cell deficiency (NKD), three encode components of CMG, Psf1 GINS1, Sld5 GINS4 and MCM4, and one encodes an auxiliary component of CMG, MCM10[86]. Interestingly, in the case of the Sld5 GINS4 mutations, compared to wild type, no significant changes in cell cycle profiles or DNA damage were detected, suggesting that NKD is caused by the loss of a role of CMG that is independent of its role in DNA replication per se[87]. Based on our findings in *C. elegans*, we speculate that a reduction in Sld5 GINS4 results in the deregulation of the expression of determinants of the NK fate that are subject to multilineage priming, resulting in a block in cell fate divergence and, hence, the absence of NK cells.

## Methods

### General *C. elegans* maintenance and strains
*C. elegans* strains were cultured and maintained according to standard protocols[88]. The Bristol N2 strain was used as wild-type strain, and the following transgenes and alleles were used in this study: LGI: *tyms-1(e2300*ts*), psf-2(t3443*ts*)* (this study); LGII: *his-9(n5357*gf*)*[42]; LGIII: *sals14* (P*lin-48*gfp)[45]; LGIV: *ced-3(n717)*[32]; LGV: *otIs356* (P*rab-3*NLS::rfp)[46]; *him-5(e1490)*[89]; *nIs396* (P*sams-s*gfp)[43]. In addition, the following multicopy transgenes and extrachromosomal arrays were used: *bcEx1302* (*psf-2*(+)) (this study), *bcEx1306* (P*psf-2*psf-2::gfp::psf-2* 3′UTR) (this study), *bcIs133* (P*toe-2*gfp)[33]. Throughout our studies, we used information and tools available on WormBase (https://wormbase.org/#012-34-5)[90,91].

### Extrachromosomal arrays generated
*bcEx1302* was generated by microinjection of a 3.6 kb genomic PCR fragment amplified from N2 Bristol using the primer psf-2_for 5′-ataaaagcgacaacgattgc-3′ and psf-2_rev 5′-aattcctttacgacttgcga-3′. 10 ng/ul of the PCR product was injected along with 100 ng/ul pRF4 into *psf-2(t3443*ts*)* mutants. Animals were incubated at 15 °C until L4 rollers were visible. Rollers were selected and shifted to 25 °C. Lines that grew at 25 °C were considered as rescue. *bcEx1306* was generated by microinjection of the plasmid pBC1695 (P*psf-2*psf-2::gfp::psf-2* 3′UTR). pBC1695 was injected (10 ng/ul) along with pRF4 (100 ng/ul) into *psf-2(t3443*ts*)* mutants. Animals were incubated at 15 °C until L4 rollers were visible. Rollers were selected and shifted to 25 °C. Lines growing at 25 °C were considered as rescue.

### EMS mutagenesis
A screen for temperature-sensitive embryonic lethal mutants using Ethyl methane sulfonate (EMS) was conducted in the laboratory of Ralf Schnabel (TU Braunschweig, Germany) using the following protocol[88]. P0s were mutagenized with 50 mM EMS at room temperature for 4 h, distributed among large (90 mm) plates and incubated overnight at 15 °C for recovery. After 24 h, mutagenized P0s were picked onto large plates (25–30 worms/plate) and incubated for 7 days at 15 °C. The F1

generation was picked onto large plates (25–30 worms each) and incubated for 7 days at 15 °C. We estimated to get ~2500–3000 F2 worms per plate without running out of food. L4 stage animals were selected from these F2 populations and singled into 96-well plates using a worm sorter (COPAS, Union Biometrica). We performed four independent screens (NC, ND, NE and NF) and singled a total of ~460.000 L4 larvae. 96-well plates were incubated at the permissive temperature (15 °C) for 7 days and then replicated with a Biomek FX (Beckman Coulter). These replica plates were incubated at the non-permissive temperature (25 °C) for 7–10 days, after which they were analyzed for lethality by eye. Compared to wells with viable animals, wells with non-viable animals still contained food. In addition, a lot of small larvae were present in these wells. Positive clones were retested manually for embryonic lethality using an eight-channel pipette for replica plating. Before phenotypic analyses were performed, positive candidates were re-tested for temperature-sensitivity a third time.

### *psf-2* cloning and temperature shift experiments with *psf-2(t3443*ts*)*
The mutant *psf-2(t3443*ts*)* was a mutant isolated in the above-described screen (NE). Using snip-SNP mapping[26], we mapped the Emb pheno-type of *t3443*ts to LGI close to the variation pkP1071 at position 23.40 cM. The whole genome sequencing was performed in Don Moermann's laboratory. The rescue experiments shown in Figs. 1 and 2 were performed using a 3778 bp PCR fragment amplified from *C. elegans* N2 strain using the primers 5′-ataaaagcgacaacgattgc-3′ and 5′-aattcctttacgacttgcga-3′ (*bcEx1302* array; Suppl Fig. 1A). The rescue experiment shown in Fig. 5 was performed using a smaller PCR fragment of 1563 bp (Suppl Fig. 1A) amplified using the primers 5′- GAA-CAGAACGATGAGCAATAC -3′ and 5′-TGGAACGTTCAACAAGTCAT-3′ (*bcEx1306* array). This smaller fragment rescues both the Emb and Ced phenotype. The smaller fragment was used to generate the plasmid pBC1695 (see General *C. elegans* maintenance and strains). Unless stated otherwise, for the analysis of the phenotype, L4 larvae were shifted to 25 °C for ~16 h before the then adult animals were dissected and embryos extracted for analysis.

### *tyms-1* cloning and temperature shift experiments with *tyms-1(e2300*ts*)*
For our analyses, we used animals homozygous for the mutation *e2300*ts, which the Schnabel lab previously defined as an allele of the gene "cib-1"[35]. We (HS and RS) have since found that *e2300*ts is a mutation in the thymidylate synthetase gene *tyms-1*. For this reason, *cib-1* was renamed "*tyms-1*". At the non-permissive temperature, embryos homozygous for *tyms-1(e2300*ts*)* presumably run out of thymidine[37,38]. This is expected to lead to a block in DNA synthesis, replication stress and general replication failure, resulting in a defect in cell division and embryonic arrest. For example, shifting *tyms-1(e2300*ts*)* L4 larvae to 25 °C results in the F1 embryos to arrest at the ~50-cell stage (Suppl. Fig. 4B)[35]. To overcome this problem, for our analyses, we shifted embryos at the one- or two-cell stage to 25 °C (rather than L4 larvae as we did for *psf-2(t3443*ts*)*). Using this temperature shift regime, compared to wild type, the cell cycle lengths of *tyms-1(e2300*ts*)* mutants progressively increase with every round of cell division similar to what we found in the case of *psf-2(t3443*ts*)* (see Fig. 1 and Suppl Fig. 3).

### Plasmid construction
DNA sequences and gene models were obtained from WormBase (WS293) and construct designs performed using SnapGene 2.8. To generate pBC1695 three fragments were amplified using N2 wild-type lysate as DNA template. The first 1331 bp long fragment was amplified using the primers 5′- gaacagaacgatgagcaatac-3′ and 5′-ctccttactcat-taaaggtgttgat-3′ and included the upstream regulatory regions and *psf-2* Exon 1 and 2 with overhangs for *egfp*. The second fragment was

894 bp long and included *egfp* with overhangs to the upstream and downstream fragments. It was amplified using the primers 5′ atcaa-caccttttaatgagtaaaggag-3′ and 5′-ggaataaaacactatttgtatagttc- 3′. The last fragment included the *psf-2* 3′UTR and the downstream regulatory region and was amplified using the primers 5′ gaactatacaaa-tagtgtttattcc-3′ and 5′-tggaacgttcaacaagtcat-3′ and had overhangs to *egfp*. All fragments were stitched via PCR stitching and cloned blunt end into the EcoRV site of pBluescript II KS. The resulting plasmid pBC1695 was sequenced for verification and injected (10 ng/ul) along with pRF4 (100 ng/ul) into *psf-2(t3443*ts*)* mutants. Animals were incubated at 15 °C until L4 rollers were visible. Rollers were selected and shifted to 25 °C. Lines growing at 25 °C were considered as rescue. The *psf-2 (RNAi)* clone was generated from the *psf-2(RNAi)* plasmid from the Vidal library. The *psf-2* cDNA fragment from the Vidal plasmid was subcloned into pBluescriptII KS(+) using EcoRV and SpeI site to generate pBC1720. All other plasmids used for RNAi were generated by amplification of cDNA fragments for each gene and were subcloned blunt end into pBluescriptII KS(+) using EcoRV site. For the *psf-3* RNAi clone a 582 bp fragment was amplified from cDNA using psf-3_cDNA_f 5′-atggctggatttgaaattc-3′ and psf-3_cDNA_r 5′-tta-taacgaaagtctcttac-3′. For the *mcm-2 (RNAi)* clone a 573 bp fragment was amplified from cDNA using mcm-2_RNAi_for 5′-gggagtagaatgga-tacatgttc-3′ and mcm-2_RNAi_rev 5′-cggagctcgatcagtactc-3′. For the *mcm-7 (RNAi)* clone a 634 fragment was amplified from exon 2 using mcm-7_Exon-2_for 5′-gacaagcaggcaatcgttg-3′ and mcm-7_Exon-2_rev 5′-ctactgggacttgctcgc-3′.

## RNA Interference
For RNAi experiments by microinjections[92], the following plasmids were used as PCR templates: pBC1720 (*psf-2(RNAi)*), pBC1721 (*psf-3(RNAi)*), pBC1719 (*mcm-7(RNAi)*), pBC1962 (*mcm-2(RNAi)*) and the following primers were used for amplification CMo24 (5′-ttgtaaaac-gacggccag-3′) and CMo25 (5′- catgattacgccaagcgc-3′) to generate PCR products, which include at the ends of the PCR product the T7 and T3 promoter. In vitro transcription was performed with Ambion Mega-script Kit T3 and T7. RNAi was performed via injection into young adult worms, which were incubated at 25 °C 3–24 h prior to recordings depending on the RNAi. For injections, Bristol N2 was used as the wild-type strain. RNAi injections were performed 24 h prior to recording in the case of *psf-2(RNAi)* and *psf-3(RNAi)*, 4 h prior to recording in the case of *mcm-7(RNAi)* and 12 h prior to recording in the case of *mcm-2(RNAi)*.

## Analysis of embryonic lethality and brood size
For the analysis of brood size and embryonic lethality, L4 larvae were picked and either maintained at the permissive temperature or shifted to the non-permissive temperature. The worms were transferred to fresh small (35 mm) plates with food twice a day until they no longer laid eggs. Shortly after transferring the worms to a fresh plate, the number of eggs laid was counted. After 24–36 h, the number of dead eggs was counted and after 24–48 h, the number of animals hatched was counted.

## 4D lineaging analysis
*C. elegans* L4 animals were grown to the adult stage overnight at 25 °C. Two- or four-cell stage embryos were harvested from the young adults and mounted on 4.5% agarose pads for DIC and fluorescence microscopy. 4D recordings were made throughout development using a Zeiss Axio Imager.M2 and Time to Live software (Caenotec), capturing 25 DIC and/or fluorescence z-stacks every 35 s at 25 °C[30,31]. Lineage analysis of the 4D recordings was performed using SIMI©BioCell software (SIMI Reality Motion Systems, http://www.simi.com)[30,31]. Cells are followed by the observer and the coordinates are recorded approximately every 2 min. The cell cleavages are assessed by marking the mother cell before

the cleavage furrow ingresses. The centers of the daughter cells are marked three frames later (105 s).

## Lineage analysis of the ABarppppp (V6R) lineage and the MSpppppp lineage
For the analysis of a specific cell (ABarp or MS) and their descendants the Software Database SIMI©BioCell was used as described above in the section "4D lineage analysis". The cells ABarp and MS are followed until ABarppppp and MSpppppp are born. These cells are the last cells in the lineage tree and differentiate into the hypodermal cell V6R (ABarppppp) and a muscle cell (MSpppppp), respectively. For the analyses, the timepoint (min) of each cell division was taken and the difference (min) (referred to as "cell cycle length") between each cell and the cell division of its corresponding mother cell was calculated.

## Lineage analysis of cells programmed to die
The 4D lineage analysis was performed as mentioned above. For the analysis of programmed cell death events, we tracked the 13 cell death events derived of the AB lineage and the MSpaapp cell death event. Cells "programmed to die" and their actual fates were followed from their births until—if possible—the engulfment of their cell corpses. If a cell died appropriately, it is indicated as "cell death" (see Fig. 2). If a cell had not died before the next round of cell division started in other cells or within a length of time corresponding to twice the cell cycle length of their mothers, it was considered inappropriately surviving and is indicated as "cell death blocked". Cells are indicated as "lost cells", if they did not fulfill either one of the two requirements described above. During our analysis we also encountered two more phenotypes in mothers of cells programmed to die. Cells are marked as "cell division of mother blocked", if a mother failed to divide and cells are marked as "mother dies", if a mother died (precocious cell death). For all cells and their descendants, cell cycles lengths were measured in minutes from birth until the next division. The cell cycle analysis of mothers of programmed cell deaths was performed from the birth of the mother cell until the mother cell divided again. Time until cell corpse formation was measured in minutes from the birth of a cell programmed to die until the formation of the "erythrocyte" stage (stage, at which the distinction between nucleus and cytoplasm is lost by DIC)[93,94].

## Lineage analysis of sisters of cells programmed to die
The 4D lineage analysis was performed as mentioned above. For the analysis of the sisters of programmed cell death events, we tracked the sisters of the 13 cell death events derived from the AB lineage. We determined whether a sister cell divided or whether its division was blocked (indicated in Fig. S4A). To that end, we tracked those cells with blocked cell divisions as far as we could at least for a length of time corresponding to twice the cell cycle length of their mothers. Cells are indicated as "lost cells", if they did not fulfill the requirement described above. Additionally, cells are indicated as "not applicable", if their mothers showed already a phenotype (either died or failed to divide).

## smRNA FISH and image analysis
smRNA FISH was performed in *C. elegans* embryos using established protocols[95]. Embryos were harvested by bleaching healthy adults and then permitted to develop in M9 buffer at 25 °C until the desired stage was reached (1 h 15 min for MSpaap cell lineage). Stellaris FISH probes (Biosearch Technologies) were designed against the mature mRNAs of *egl-1* and *ced-3*. The *egl-1* probe set contained 23 TAMRA-labeled oligonucleotides and was used at a working concentration of 250 nM in Hybridization Buffer[95]. The *ced-3* probe set contained 48 Quasar670-labeled oligonucleotides and was used at 500 nM. Image stacks were obtained using Leica LAS AF software and a Leica TCS SP5 II confocal microscope with a 63× oil immersion lens and a z-spacing of 500 nm to capture diffraction-limited mRNA spots over several z-slices. Laser intensity was set to 10% to minimize bleaching. Image analysis was

performed using Fiji 2.3 software[96]. The pipeline used to quantify the mRNA copy number in a cell of interest was outlined previously[23]. Briefly, a three-dimensional region of interest (ROI) was defined for the cell of interest as a subset of cropped z-slices that fully contained the cell. The total smRNA FISH signal intensity ($SI_{Total}$) contained within this ROI was determined by summing all z-slices and measuring the total signal in the resulting z-projection. Next, background signal was subtracted by determining the total smRNA FISH signal intensity for three neighboring regions of the same size without visible mRNA signal, then subtracting their average signal intensity ($SI_{Bkgd}$) from that of the ROI. Finally, the mRNA copy number was calculated by dividing the total signal intensity of the ROI by the average intensity of a single diffraction-limited mRNA spot ($SI_{Spot}$), or generally, the mRNA copy number in a cell was calculated as ($SI_{Total} - SI_{Bkgd}$)/$SI_{Spot}$. For presentation, maximum intensity z-projection images were smoothed (Gaussian blur; radius, 1.5). To calculate cellular concentrations of mRNA in the MSpaap and MSpaapp cells, first the average volumes of these two cells were determined by assuming sphericity and measuring their average diameters from confocal image stacks. Then, the mRNA copy number in each cell of interest was divided by the average volume of that cell. Finally, the total number of embryonic nuclei was counted so that the cell of interest could be mapped to a specific developmental timepoint. This nuclei count was performed using Multiview Reconstruction software[97] to detect interest points in the DAPI channel, with manual corrections as required. To generate a developmental time-course plot of mRNA concentration, mRNA copy numbers were first ordered chronologically by increasing embryonic nuclei count. Next, a centered running average of order 5 was applied to both embryonic nuclei count and mRNA copy number data, effectively smoothing the resulting plot. Finally, shaded areas representing the SEM of averaged data points were added to the plot.

### Quantification of QL.pp death
QL.pp death was analyzed using the P_{toe-2}gfp (bcIs133) transgene[33], which labels cells of the Q lineage. Within the QL.p lineage, L2 larvae of wild-type animals contain two GFP-positive PVM and SDQL neurons, which are the daughters of the surviving sister QL.pa. In cell death mutants, up to two extra GFP-positive cells can be seen, which are undead QL.pp cells[33]. For quantification, gravid wild-type adults maintained at 25 °C were allowed to lay eggs at 25 °C for 1 h. The adults were removed, and the eggs laid were incubated at 25 °C for 24 h, until they reached the L2 stage. The animals on the plate were washed off with 2 mM levamisole solution in MPEG and collected by centrifugation at 400 g for 1 min. 5 μL of the pelleted worms was then mounted on a 2% agarose pad on a glass slide and an 18 × 18 mm coverslip (#1.5 thickness) was added on top. The number of GFP-positive cells was counted using the Zeiss Axio Imager M2 equipped with the ZEN 2.6 pro (Blue edition) software with a 100×/1.3 NA oil-immersion objective lens[33]. Only those worms were assessed where the PVM and SDQL (QL.pa daughter cells) had formed obvious neurite extensions. For quantification of psf-2(t3443ts) animals at the non-permissive temperature, the strain was maintained at 15 °C, and gravid adults were allowed to lay eggs at 25 °C for 2 h. The adults were removed, and eggs laid were further incubated at 25 °C for 28–30 h until they reached L2 stage. The larvae were mounted and scored as described above for wild-type animals. In the psf-2(t3443ts) background, only one GFP-positive cell with neurite extensions was observed in some cases, indicating a block in cell division in QL.p or QL.pa.

### Analysis of MI fate
MI fate was analyzed using the P_{sams-5}gfp (nIs396) transgene, which is expressed exclusively in the MI neuron in wild-type animals[42,43]. Wild-type animals (15 °C and 25 °C), his-9(n5357gf) animals (15 °C and 20 °C) and psf-2(t3443ts) animals (15 °C) were grown at the respective temperatures for at least two generations before assaying the MI fate. L4

larvae were mounted on 2% agarose pads containing 25 mM sodium azide in M9 buffer and an 18 × 18 mm coverslip (#1.5 thickness) was added on top. The number of GFP-positive cells was counted in the anterior pharynx using the Zeiss Axio Imager M2 equipped with the ZEN 2.6 pro (Blue edition) software with a 100×/1.3 NA oil-immersion objective lens. For scoring psf-2(t3443ts) animals at the non-permissive temperature of 25 °C, the strain was maintained at 15 °C and then gravid adults were allowed to lay eggs at 25 °C for 6 h. The adults were removed, and the eggs laid were shifted back to permissive temperature of 15 °C for 5 days. On the 5th day, L4 larvae were mounted and GFP-positive cells were counted in the anterior pharynx as described above.

### Analysis of AMso division and MCM fate
psf-2(t3443ts) and wild-type animals were grown at the permissive temperature (15 °C). Synchronized populations of L1 larvae, obtained by hypochlorite treatment and hatching in M9 buffer, were shifted to the non-permissive temperature (25 °C). The AMso is born 310 min post-fertilization, thus the L1 shift ensures that only the asymmetric division to produce the MCM is affected. 1-day adult male animals were scored for AMso division and the presence of the MCM neuron. To monitor the division, the saIs14 (P_{lin-48}gfp) transgene was used[45]. In wild-type animals, the saIs14 transgene is expressed in the AMso mother (ABpli/rpaapapa) starting with its birth during embryogenesis. After the postembryonic division (~32–36 h post-L1 arrest), it is continuously expressed in the anterior AMso daughter (ABpl/rpaapapaa), temporarily retained in the posterior daughter cell (ABpl/rpaapapap) and gradually lost as it differentiates into the MCM neuron. To assess neuronal identity, the otIs356 (P_{rab-3}NLS::rfp) panneuronal reporter transgene was used[46]. Cells per side were quantified, with wild-type animals having two lin-48 and one rab-3 expressing cell per side. psf-2(t3443ts) animals maintained at the permissive temperature (15 °C) were scored as controls. Animals carrying the rescuing transgene were maintained and scored at the non-permissive temperature (25 °C). Animals considered wild-type carry him-5(e1490).

### Fluorescence Imaging
For the MI fate analysis, images of anesthetized worms were acquired on the Zeiss Axio Imager M2 with a 100×/1.3 NA oil-immersion objective lens. A stack that included the entire anterior pharynx of the larvae was acquired in the GFP and DIC channel with step-size of 0.5 μm. Using Fiji (ImageJ), a maximum intensity projection of the GFP channel was generated (to ensure all GFP-positive cells in the anterior pharynx were included in the image). A central slice for the DIC channel was chosen that accurately represented the GFP-positive cell. The GFP and DIC images were merged and used for Fig. 5B. For AMso and MCM imaging, worms were anesthetized using 50 mM sodium azide and mounted on 5% agarose pads on glass slides. Images were acquired on a Zeiss AxioImager with a 40×/1.3 NA oil-immersion objective lens coupled to a 2.5 zoom, using a Zeiss Colibri LED fluorescent light source and custom TimeToLive multi-channel recording software (Caenotec). Representative images are shown following maximum intensity projections of 2–10 1 mm z-stack slices edited using Fiji.

### Reporting summary
Further information on research design is available in the Nature Portfolio Reporting Summary linked to this article.

## Data availability
All data and material used in this manuscript are available and can be requested from the corresponding authors. The raw data generated in this study are provided in the Source Data file. Source data are provided with this paper.

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

## Acknowledgements

We thank members of the Conradt, Schnabel, Lambie and Poole labs for discussions and Stéphane Rolland for comments on the manuscript. We thank C. Struck and L. McGuinness for their excellent technical support. Some strains were provided by the *Caenorhabditis* Genetics Center (CGC), which is funded by NIH Office of Research Infrastructure Programs (P40 OD010440). We thank Don Moerman and the *C. elegans* Knockout Facility in Vancouver B.C. Canada (https://www.zoology.ubc.ca/~dgmweb/) for providing the annotated single nucleotide variants and indels in the *psf-2(t3443)* strain. R.S. was supported by a Postgraduate Scholarship - Doctoral (PGS D) from the Natural Sciences and Engineering Research Council of Canada (NSERC). This work was supported by UCL (Division of Biosciences, UCL LSM Capital Equipment Fund to B.C.), by a previous Wellcome Senior Research Fellowship (207483/Z/17/Z) and a current BBSRC Responsive Mode Grant (BB/X00208X/1) to R.J.P, a Korean Institute for Basic Science Grant (IBS-R022-D1) to N.M. and a Wolfson Fellowship from the Royal Society (RSWF\R1\180008) and BBSRC Responsive Mode Grants (BB/V007572/1 and BB/V015648/1) to B.C.

## Author contributions

Experiments were performed by N.M., Ryan Sherrard, A.S., C.L.F., and H.S. All authors (N.M., Ryan Sherrard, A.S., C.L.F., H.S., E.J.L., R.J.P., Ralf Schnabel, B.C.) participated in designing experiments, data analysis, and data interpretation. All authors also provided input and revisions to successive drafts of the entire manuscript. B.C. managed the overall project and obtained funding.

## Competing interests

The authors declare no competing interests.
