## [Peer Review file · Nature Communications]

The replicative helicase CMG is required for the divergence of cell fates during asymmetric cell division in vivo

Corresponding Author: Professor Barbara Conradt

Version 0:

Reviewer comments:

Reviewer #1

(Remarks to the Author)

This is a very interesting paper looking at the role of a DNA replication helicase PSF2 in the initiation of cell death in *C. elegans*. The authors demonstrate that a conditional mutation in the helicase results in cell cycle prolongation that is separate from a block in the initiation of many cell death events they examined. They go on to show that the cell death defect likely arises because of a reduction in expression of a BH3-only gene, *egl-1*.

Mechanisms that initiate developmental cell death are not fully understood. *C. elegans* offers a great system to analyze this question as cell death events occur reproducibly at defined times and lineages. There has been speculation that cell cycle can influence fate determination, also in the context of cell death. This paper reveals that even though the cell cycle per se may not be relevant, a cell cycle component is. The experiments are appropriate, the quality of the data is outstanding, and the conclusions are warranted. The authors should be congratulated for the rigor of their studies.

I have 2 points which the authors may wish to consider:

1. While the replication complex appears not relevant for the cell death effects, it remains possible that the helicase activity is important during mRNA synthesis, to unwind DNA to allow RNA production. It may be, therefore, useful to test if specifically mutating critical residues for helicase function also affects cell death. Alternatively, as this experiment may be somewhat challenging if an early lethality of embryos is observed, could the authors express another DNA helicase and see if they can get rescue?
2. For the expression studies, while the FISH experiments are beautiful and convincing, it would be nice to see confirmation using another method. Is there a reason not to examine the *egl-1* gfp reporter that the authors discuss in the paper?

Reviewer #2

(Remarks to the Author)

The manuscript by Memar et al presents a characterization of the effects of a temperature sensitive mutation in the *psf-2* gene, which the authors show encodes a component of the conserved CMG helicase required for DNA replication in many organisms. The authors show that both this mutation and RNAi of *psf-2* cause delayed cell cycles consistent with a defect in DNA replication. Interestingly though, they also identify a role for PSF-2 in cell death. Specifically, cell death, which is a programmed fate of resulting from asymmetric division in *C. elegans*, is blocked and/or delayed in many cells in *psf-2*(ts) mutants. The authors provide a nice characterization of cell cycle relative to the cell death phenotype which convincingly shows there is no correlation between those two phenotypes. Further, they find that RNAi depletion of other components of CMG cause both cell cycle and cell death phenotypes similar to those of the *psf-2*(ts) mutation. In contrast, the loss of a gene required for thymidine synthesis and thus DNA replication, *tym-1*, causes a longer cell cycle delay without concomitant cell death defects. Thus, the authors conclude that controlling cell death fate is a unique function of CMG separable from its role in DNA replication. The authors are able to link the cell death defect of *psf-2* mutants to changes in the transcription of the cell death promoting gene *egl-1*. Specifically, although levels of *egl-1* transcript in the mother cell are at similar low levels as in controls (a presumed "poised for transcription state"), *psf-2* mutants do not show the normal increase in *egl-1* transcripts that causes cell to die. Thus, the study shows that CMG is required for the asymmetric transcription of *egl-1* that results in an

asymmetry of cell fates. Overall, the majority of the experiments are well done and rigorous and the conclusions are supported by the evidence, with some issues/exceptions noted below. The authors have uncovered an exciting and novel role for the CMG complex in cell fate and propose that CMG acts epigenetically to control *egl-1* transcription. This study should be of interest to a variety of researchers from different areas.

Major Concerns- issues that affect conclusions:

1) Fig. 4 and the corresponding results section clearly show that the mother cell in *psf-2* mutants has the same levels of *egl-1* transcripts as in controls, and that daughter cell that should die doesn't show the increase in *egl-1* transcripts. But I found it somewhat strange that the levels in the sister cell- that normally doesn't die- were not examined. Does it still downregulate *egl-1*? Why were no other markers for cell fate examined (especially in light of looking at markers for other lineages in the last section). Such analysis would give more insight into how the asymmetric division has been affected. Is the fate completely normal, or some type of hybrid? And for the non-expert, it would be helpful to clearly state that the fates of the asymmetric division have not been switched (which does occur in some ACD mutants); I am assuming they are not switched because no new "X" for cell death appeared in the lineage but I don't believe this was explicitly stated.

2) The authors did seek to show other asymmetric divisions/ fates are affected and make the conclusion that CMG is not specific to affecting cell death lineages. I found this to be the weakest section of the paper for several reasons. First, I didn't feel there was a clear rationale for why these two fates – M1 and AMso - were chosen. Reading between the lines later, I believe it is because the authors are trying to link CMG to *his-9* and histones with an effect on M1. But this isn't clearly laid out to start with, and in the end the defect in M1 in *psf-2* mutants is very low penetrance. Thus, I do not think the conclusion (in the Discussion) that CMG acts in the same process with *HIS-9* is well supported by this finding. If these two act in the same mechanism, wouldn't one expect the *his-9* mutant to affect the same cell death lineages described earlier in the paper? But this wasn't tested. The Discussion section incorporating this data also seems very lengthy given the tenuousness of these results- I think the authors could speculate on a role for *PSF-2* in histone/nucleosome remodeling much more concisely.

Further, in this section, no detailed analysis was done as for the prior lineages – ie it is not shown that the defects in the M1 asymmetric division, or the AMso one – are not indirect effects of the replication delay. Thus overall, I find the conclusions here less supported. I am not asking for that more thorough analysis necessarily, but care must be taken about the strength of such conclusions. And again I think additional information about the MS and AB lineage fates and/or a test of whether *his-9* affects those lineages would strengthen the study.

3) Picky but important for accuracy: Line 176. The statement that *psf-2* affects cell death "in all cell death lineages" implies in every single one of the embryo and post-embryonically, which hasn't been analyzed here. Did you mean "in any of the various lineages of the 1st wave of cell death" ?

And then, from my understanding this along with the experiment on post-embryonic death does suggest that *psf-2(ts)* results in a "general" block, but it would help the reader to clarify perhaps the types of divisions of this first wave...e.g. are they all divisions that normally result in neuron and a cell death, or doe these represent a variety of different fates vs cell death. Even most *C. elegans* readers won't know which cell fates may be in these specific MS and AB sublineages.

4) Picky but important for accuracy: Several places it is stated that role of *PSF-2* in cell death fate is independent or separate (several different wordings used) of its DNA helicase/unwinding activity. Because this isn't being tested here in *C. elegans*, and it is unclear what this *psf-2(ts)* mutation does to the protein's activity, I think it is more accurate to use wording that this role is separable from its role in DNA replication (and could add, thus presumably DNA helicase activity)

Minor issues/suggestions to improve clarity or readability

Line 47 - "whether cell division itself..." I don't think a good choice of words because that encompasses asymmetric segregation of components, mitotic spindle control etc and I don't think what the authors intend

Line 99 -For non *C. elegans*, even non-*C. elegans* embryo experts, it would be helpful to give the reader an idea of the normal number of cells at the pre-morphogenetic stage, for comparison later when *psf-2(RNAi)* causes arrest at 50-cell stage (I know you didn't count, but you could state the wt # at this stage),.

Line 132 section -Why this lineage? Overall the paper is very clearly written and flows nicely. However, since the abstract and Intro are about ACD and cell death, but it takes a long time to get to anything to do with those topics, I found myself wondering how you found this mutant in the first place. Were you examining lineage delays, or looking for new cell death mutants? And then specifically here starting with line 132, does this lineage have cell death? I didn't see any X's in the figure so assuming not and they appear different sublineages than in Fig. 2. Again, especially for a non-*C. elegans* person all the cell labels may be daunting and so stating more explicitly why these cells were examined could help with readability.

Line 148 – It would help if you emphasized that you did longer RNAi of 48hr, which gave a stronger phenotype of arrest of 50cell. This is indicated in parentheses, but I had to go back and hunt for it and so missed the point the first time.

Line 186 – "This indicates that in 38% of..." The sentence does not follow from the data provided in the prior sentence or easily from the graph. An average time was given, not the number of cells that divided within the normal wt range vs out of that range, which is what the sentence seems to be indicating.

Version 1:

Reviewer comments:

Reviewer #1

(Remarks to the Author)

I have read the authors' responses to the comments of the reviewers. While they acknowledge the possibility of the helicase-transcription model I proposed, they do not perform any of the studies I suggested. I would recommend, therefore, that the authors at least acknowledge in the manuscript the possibility of this alternative model. They have already changed some of the language regarding the separate replication and cell death mechanisms in response to the other reviewer's comments, and this specific model could also be added to those changes. Otherwise, I have no other concerns.

Reviewer #2

(Remarks to the Author)

The authors have very nicely addressed all my major and minor comments. I have no further concerns.

REVIEWER COMMENTS

Reviewer #1 (Remarks to the Author):

This is a very interesting paper looking at the role of a DNA replication helicase PSF2 in the initiation of cell death in *C. elegans*. The authors demonstrate that a conditional mutation in the helicase results in cell cycle prolongation that is separate from a block in the initiation of many cell death events they examined. They go on to show that the cell death defect likely arises because of a reduction in expression of a BH3-only gene, *egl-1*.

Mechanisms that initiate developmental cell death are not fully understood. *C. elegans* offers a great system to analyze this question as cell death events occur reproducibly at defined times and lineages. There has been speculation that cell cycle can influence fate determination, also in the context of cell death. This paper reveals that even though the cell cycle per se may not be relevant, a cell cycle component is. The experiments are appropriate, the quality of the data is outstanding, and the conclusions are warranted. The authors should be congratulated for the rigor of their studies.

I have 2 points which the authors may wish to consider:

1. While the replication complex appears not relevant for the cell death effects, it remains possible that the helicase activity is important during mRNA synthesis, to unwind DNA to allow RNA production. It may be, therefore, useful to test if specifically mutating critical residues for helicase function also affects cell death. Alternatively, as this experiment may be somewhat challenging if an early lethality of embryos is observed, could the authors express another DNA helicase and see if they can get rescue?

Memar et al

The replicative helicase complex CMG (Cdc45-MCM-GINS) has two enzymatic activities: it has DNA helicase activity (provided by the MCM subcomplex), and it has histone chaperone activity (MCM-2, a component of the MCM subcomplex, binds to histones and provides the histone chaperone activity). The mutation *psf-2(t3443ts)* (which may affect the stability of the entire CMG complex) causes increasing cell cycle lengths and a cell death defect. Our results suggest that these two defects represent separate (rather than functionally connected) defects. We propose that the cell cycle defect is the result of compromised CMG DNA helicase activity, and this is mainly based on previous work by the replication and cell cycle fields. On the other hand, we propose that the cell death defect is the result of compromised CMG histone chaperone activity. This is based on the recent findings by the labs of Zhiguo Zhang, Anja Groth and Haiyun Gan (citations 48-50 in the manuscript) that the loss of CMG histone chaperone activity in mouse ESCs causes a defect in the expression of key differentiation genes and our finding that *psf-2(t3443ts)* abolishes the increase in expression of such a key differentiation gene, *egl-1*.

The reviewer now suggests an alternative hypothesis that - by unwinding DNA at the *egl-1* locus thereby permitting RNA production - the helicase activity of CMG might also be required for *egl-1* transcriptional upregulation. Based on this alternative model, the reviewer asks whether expressing a different helicase might rescue the cell fate defect in *psf-2(t3443ts)* mutants. We consider this model very interesting but challenging to address for the following reasons. First, we agree with the reviewer, mutating critical residues in the MCM subcomplex required for DNA helicase activity is expected to block cell division, causing early lethality. For this reason, we would not be able to assess the impact on cell death. Second, the *C. elegans* genome encodes a total of 20 DNA helicases; however, it is unclear which - if any - of these helicases might be

able to function in the appropriate context. For this reason, we consider the proposed experiment currently not feasible.

2. For the expression studies, while the FISH experiments are beautiful and convincing, it would be nice to see confirmation using another method. Is there a reason not to examine the *egl-1 gfp* reporter that the authors discuss in the paper?

Memar et al

We agree with the reviewer that confirming our results using another method would make our data even more convincing. Unfortunately, the *egl-1* transcriptional *gfp* reporter mentioned in the manuscript does not give us the temporal resolution necessary to look at the very dynamic changes in *egl-1* expression during asymmetric mother cell division. Because of the time it takes GFP to assemble and to become fluorescent and the fact that apoptotic cells die very rapidly (within 20-30 min post-cytokinesis), in a wild-type background, we have so far failed to detect any GFP signal from this reporter in the cell death lineages that we examined for the current study. (The work we mention in the manuscript using the *gfp* reporter was done in a *ced-3* caspase mutant background, in which cell death is blocked.) In the future, we plan to establish the MS2 system for monitoring *egl-1* transcriptional activity in embryos. If successful, this would allow us to monitor *egl-1* transcription in real time (rather than in fixed embryos).

Reviewer #2 (Remarks to the Author):

The manuscript by Memar et al presents a characterization of the effects of a temperature sensitive mutation in the *psf-2* gene, which the authors show encodes a component of the conserved CMG helicase required for DNA replication in many organisms. The authors show that both this mutation and RNAi of *psf-2* cause delayed cell cycles consistent with a defect in DNA replication. Interestingly though, they also identify a role for PSF-2 in cell death. Specifically, cell death, which is a programmed fate of resulting from asymmetric division in *C. elegans*, is blocked and/or delayed in many cells in *psf-2(ts)* mutants. The authors provide a nice characterization of cell cycle relative to the cell death phenotype which convincingly shows there is no correlation between those two phenotypes. Further, they find that RNAi depletion of other components of CMG cause both cell cycle and cell death phenotypes similar to those of the *psf-2(ts)* mutation. In contrast, the loss of a gene required for thymidine synthesis and thus DNA replication, *tyms-1*, causes a longer cell cycle delay without concomitant cell death defects. Thus, the authors conclude that controlling cell death fate is a unique function of CMG separable from its role in DNA replication. The authors are able to link the cell death defect of *psf-2* mutants to changes in the transcription of the cell death promoting gene *egl-1*. Specifically, although levels of *egl-1* transcript in the mother cell are at similar low levels as in controls (a presumed "poised for transcription state"), *psf-2* mutants do not show the normal increase in *egl-1* transcripts that causes cell to die. Thus, the study shows that CMG is required for the asymmetric transcription of *egl-1* that results in an asymmetry of cell fates. Overall, the majority of the experiments are well done and rigorous and the conclusions are supported by the evidence, with some issues/exceptions noted below. The authors have uncovered an exciting and novel role for the CMG complex in cell fate and propose that CMG acts epigenetically to control *egl-1* transcription. This study should be of interest to a variety of researchers from different areas.

Major Concerns- issues that affect conclusions:

1) Fig. 4 and the corresponding results section clearly show that the mother cell in *psf-2* mutants

has the same levels of egl-1 transcripts as in controls, and that daughter cell that should die doesn't show the increase in egl-1 transcripts. But I found it somewhat strange that the levels in the sister cell- that normally doesn't die- were not examined. Does it still downregulate egl-1?

Memar et al

As outlined in more detail below, our data indicate that in psf-2(t3443ts) mutants, egl-1 is still downregulated in the surviving sister. We thank the reviewer for making this comment and for pointing out that this was not made clear in the original manuscript.

The reason why we did not report egl-1 mRNA levels in the surviving sister in Figure 4 (smRNA FISH experiments) is the following. Since the embryos are fixed and since in eight out of 11 embryos egl-1 mRNA upregulation is no longer observed in the normally-destined-to-die MSpaapp cell, it is difficult to unequivocally identify the unwanted daughter and its sister cell. Therefore, as described in the Results section on page 16, we averaged the smRNA FISH signal of four cells within the area where those two cells are normally located in order to estimate the value for the MSpaapp cell:

Page 16

“Indeed, in eight out of 11 embryos (about 73%), essentially no egl-1 mRNA was detectable in MSpaapp (Fig. 4B, psf-2(t3443ts), egl-1). (In the case of these eight embryos, we measured egl-1 mRNA in four cells at the position where MSpaapp is usually located, determined the average egl-1 mRNA copy number of these four cells and used this as a value for MSpaapp.)”

However, in these eight embryos, we did not see increased levels of egl-1 mRNA in any adjacent cells. For this reason, we are confident that the fates are not switched, and that egl-1 is still downregulated in the surviving sister. We have added the following statement on page 16, 17 of the Result section to clarify this:

Page 16, 17

“Importantly, in these eight embryos, we did not detect neighboring cells with increased levels of egl-1 mRNAs. This supports the notion that the absence of egl-1 mRNAs in MSpaapp in psf-2(t3443ts) animals is the result of the loss of egl-1 expression in MSpaapp rather than a switch in daughter cell fates and the concomitant ‘gain of egl-1 expression’ in MSpaapp’s sister cell MSpaapa (see Fig. 4B).”

As outlined in more detail below, in response to additional comments made by the reviewer, we have also gone back to our psf-2(t3443ts) recordings and lineaged the surviving sisters of the 13 AB-derived 1st wave cell deaths, which we analysed for the data presented in Figure 2 and 3. Our results support the notion that cells that normally die do not simply swap fates with their surviving sisters (see below).

Why were no other markers for cell fate examined (especially in light of looking at markers for other lineages in the last section). Such analysis would give more insight into how the asymmetric division has been affected. Is the fate completely normal, or some type of hybrid? And for the non-expert, it would be helpful to clearly state that the fates of the asymmetric division have not been switched (which does occur in some ACD mutants); I am assuming they are not switched because no new “X” for cell death appeared in the lineage but I don't believe this was explicitly stated.

Memar et al

We thank the reviewer for pointing out that this was not clearly stated in the manuscript. Yes, the fates are indeed not switched. As stated above, we have now added a statement on page 16, 17 of the revised manuscript that clarifies that the fates of MSpaapp and MSpaapa are not switched (Figure 4, smRNA FISH data).

In addition, to obtain more evidence that the fates are not switched, we have gone back to our *psf-2(t3443ts)* recordings and analysed the fates of the sisters of the 13 AB-derived 1st wave cell deaths. Our results indicate that the sisters do not die but continue on their normal developmental trajectory. We have included these new data in the revised manuscript (new **Supp. Figure 4**) and added the following to the Results section on page 10:

Page 10

*“Finally, the 1st wave cell deaths derived from the AB blastomere are generated through asymmetric cell divisions. To rule out that *psf-2(t3443ts)* causes a switch of daughter cell fates rather than a block in cell death, we analyzed the fates of the sisters of the 13 AB-derived 1st wave cell deaths in four *psf-2(t3443ts)* embryos (**Suppl. Fig. 4**). In wild type, 100% of the sister cells divide and give rise to two daughter cells. In *psf-2(t3443ts)* animals, 66% of the sister cells divide and give rise to two daughter cells. In the remaining 34.2%, cell division is blocked (**Suppl. Fig. 4A, B**). Importantly, 0% of the sister cells inappropriately die (**Suppl. Fig. 4B**), confirming that *psf-2(t3443ts)* does not cause a switch of daughter cell fates. These results are exemplified in **Supplementary Figure 4C** for the sister of the 1st wave cell death ABalaapapa referred to as CD#1. In three out of four *psf-2(t3443ts)* embryos (embryos #1-3), the sister of CD#1 (ABalaapapp) divides and in the remaining embryo (embryo #4), it fails to divide.”*

Concerning cell fate markers, we were not able to assess other cell fate markers in the case of the 1st wave cell deaths, because the surviving sisters do not differentiate right away and therefore do not yet express cell fate markers. However, with the additional lineaging data about the fates of the surviving sisters (new **Supp. Figure 4**), we hope we have addressed the reviewer's concerns regarding the fates of the surviving sister cells.

Finally, as outlined in more detail below, one reason why we decided to look at the division of AMso is that for this lineage, cell fate markers are available for both daughter cells. Indeed, we used both markers but failed to describe this appropriately in the original manuscript (see below). Specifically, as shown in Figure 5e, in wild type, 100% of the AMso divide to give rise to two daughter cells, AMso and MCM. AMso and MCM both are positive for the glial marker *lin-48* but only MCM is positive for the neuronal marker *rab-3*. In *psf-2(t3443ts)* animals (at 25°C), 79% of the AMso divide to give rise to two cells both of which are positive for *lin-48*. Among these 79%, AMso divisions that generate a *rab-3* positive MCM daughter cell account for 8%, AMso divisions that generate a *rab-3* negative MCM daughter cell account for 68%, and AMso divisions that generate two *rab-3* positive cells account for the remaining 3%. Importantly, in none of the AMso divisions analysed did we observe that the fates of AMso and MCM were switched, i.e. that the anterior cell was *rab-3*-positive.

To clarify this, we added the following statement on page 19, 20 of the Result section:

Page 19, 20

“Importantly, we never observed a switching of the glial and neuronal fates i.e. that the anterior daughter adopts the neuronal fate and expresses the $P_{rab-3NLS}::rfp$ reporter. This supports the

notion that in the AMso lineage, psf-2(t3443ts) causes the loss of the neuronal fate rather than a switching of daughter cell fates.”

2) The authors did seek to show other asymmetric divisions/ fates are affected and make the conclusion that CMG is not specific to affecting cell death lineages. I found this to be the weakest section of the paper for several reasons. First, I didn't feel there was a clear rationale for why these two fates – M1 and AMso - were chosen. Reading between the lines later, I believe it is because the authors are trying to link CMG to his-9 and histones with an effect on M1. But this isn't clearly laid out to start with, and in the end the defect in M1 in psf-2 mutants is very low penetrance. Thus, I do not think the conclusion (in the Discussion) that CMG acts in the same process with HIS-9 is well supported by this finding. If these two act in the same mechanism, wouldn't one expect the his-9 mutant to affect the same cell death lineages described earlier in the paper? But this wasn't tested. The Discussion section incorporating this data also seems very lengthy given the tenuousness of these results- I think the authors could speculate on a role for PSF-2 in histone/nucleosome remodeling much more concisely. Further, in this section, no detailed analysis was done as for the prior lineages – ie it is not shown that the defects in the M1 asymmetric division, or the AMso one – are not indirect effects of the replication delay. Thus overall, I find the conclusions here less supported. I am not asking for that more thorough analysis necessarily, but care must be taken about the strength of such conclusions. And again I think additional information about the MS and AB lineage fates and/or a test of whether his-9 affects those lineages would strengthen the study.

Memar et al

Concerning the **MI fate decision**, we decided to analyse this cell fate decision because it is a cell fate decision that does not generate a cell death and because – as the reviewer suspected – we wanted to connect our data with that previously published by Horvitz, Stillman and co-workers (Nakano et al, 2011), who proposed that this cell fate decision is dependent on epigenetically distinct chromatids that are generated in the mother cell through unequal replication-coupled nucleosome assembly and that are non-randomly segregated into the daughter cells. We are now stating the rationale for choosing this cell fate decision in the Result section on page 18, 19.

Page 18, 19

“ABaraappaa ('MI mother cell') divides during the 10th round of cell division and generates an anterior daughter, ABaraappaaa, which differentiates into the pharyngeal motor neuron/interneuron MI, and a posterior daughter cell, ABaraappaap, which differentiates into the pharyngeal marginal cell m1DR ('MI/m1DR decision')⁹. A cold-sensitive (cs) gain-of-function (gf) mutation of the gene his-9, n5357gf, one of 14 genes in the C. elegans genome that encode replication-dependent histone H3, has been proposed to interfere with H3-H3 interactions and, hence, the formation of H3-H4 tetramers and nucleosome assembly⁴². Importantly, it was previously shown that in his-9(n5357gf) animals, instead of differentiating into the MI neuron, ABaraappaaa adopts the fate of the pharyngeal epithelial cell e3D⁴². Based on this, Horvitz, Stillman and co-workers proposed that the ability of ABaraappaa to adopt the MI fate is dependent on epigenetically distinct sister chromatids that are generated in the MI mother cell through unequal nucleosome transfer during replication and that are selectively segregated into the two daughter cells⁴². To determine whether the role of psf-2 GINS2 in cell fate divergence extends to cell fate decisions that, first, do not generate a daughter that dies and, second, have been proposed to involve epigenetically distinct chromatids generated through replication-coupled nucleosome assembly, we, next, analyzed the MI/m1DR decision.”

We did analyse the effect of the published *his-9(n5357gf)* mutation on the 1st wave of cell death; however, we did not see an obvious phenotype (n=19 cell deaths analysed). *his-9* is one of a total of 14 genes in the *C. elegans* genome that encode replication-dependent Histone H3 proteins. Their expression patterns are not known in detail, but we speculate that *his-9* most likely is not a Histone H3 gene that is prominently expressed during the 1st wave of cell death. Indeed, in their 2011 paper, Horvitz, Stillman and co-workers demonstrated that introducing the identical change found in the *his-9* gene in *n5357* animals (resulting in H113D) into other replication-dependent Histone H3 genes, caused much weaker phenotypes with respect to the MI fate decision (see part of Figure 4B from Nakano et al, 2011 below). For this reason, we decided to look at the MI fate decision to see whether *psf-2(t3443ts)* animals had a phenotype. Since the MI fate decision occurs later during embryonic development, it was difficult to establish a temperature shift protocol that allows the embryo to develop far enough for this cell fate decision to take place but still reduce *psf-2* function to a level that may be sufficiently low to compromise its histone chaperone activity. We think that this contributed to the relatively weak MI phenotype we observed in *psf-2(t3443ts)* animals.

Genomic clone	% MI transformation
his-13(+)	0 (n = 130)
his-13(H113D)	13 (n = 127)
his-32(+)	0 (n = 132)
his-32(H113D)	4 (n = 141)
his-2(+)	0 (n = 169)
his-2(H113D)	2 (n = 125)

Figure 4B, Nakano et al, 2011.

In the revised manuscript, we have toned down the conclusions from this experiment in the Results section (page 19) as well as the Discussion (page 25).

Page 19:

*“Therefore, reducing *psf-2* GINS2 function can impact the MI/mIDR decision – albeit at low penetrance - and, hence, a process that has been proposed to involve epigenetically distinct chromatids generated through unequal replication-coupled nucleosome assembly.”*

Page 25:

*“Second, the mutation *his-9(n5357gf)*, which interferes with the formation of H3-H4 tetramers, results in the loss of the MI fate⁴² (see **Figure 5**). We show that reducing CMG also results in the loss of the MI fate albeit at low penetrance.”*

In the case of the **AMso/MCM decision**, the rationale for looking at it was the following. First, it is a cell fate decision that occurs post-embryonically. Second, analysing the AMso division confirms that *psf-2* is acting in the mother (AMso) rather than other progenitors as the temperature shifts in the *psf-2(t3443ts)* background are not performed until *after* the birth of the AMso in the embryo. This was not clear from our analyses of the 1st wave cell deaths as the embryos examined in this case were maintained at 25°C at all times. Third, we have compatible cell fate markers available to us (green and r

ed) that allow us to analyse the fates of the daughter cells in one and the same animal. This allowed us to demonstrate that the daughter cell fates are not switched. We are now stating the rationale for choosing this cell fate decision in the Results section on page 19, 20.

Pages 19, 20:

“The AMso cells (ABpl/rpaapapa) are a pair of bilaterally symmetrical glial cells that are born in the embryo. In males, each of the AMso cells divides asymmetrically during post-embryonic development to generate another glial cell (ABpl/rpaapapaa referred to as ‘AMso’) and a MCM neuron (ABpl/rpaapapap) (‘AMso/MCM decision’)⁴⁴. Next, we decided to analyse the AMso/MCM decision to determine whether the role of psf-2 GINS2 in cell fate divergence, first, extends to cell fate decisions that occur during post-embryonic development and, second, is required in mother cells. In addition, the availability of compatible reporters allowed us to simultaneously monitor both daughter cell fates.”

Concerning the MS and AB lineage fates, as stated above (Reviewer’s point 1), the surviving sisters of all 14 1st wave cell deaths do not differentiate but divide again and some of their daughters go through one more round of division. Depending on the specific lineage, these daughters and granddaughters differentiate into different types of cells (11 neurons, 1 valve cell, 1 marginal cell, 2 hypodermal cells (hyp8/9), 2 sheet and 2 socket cells, 1 excretory cell and 1 epithelial cell).

3) Picky but important for accuracy: Line 176. The statement that psf-2 affects cell death “in all cell death lineages” implies in every single one of the embryo and post-embryonically, which hasn’t been analyzed here. Did you mean “in any of the various lineages of the 1st wave of cell death” ?

Memar et al

Indeed, we meant in all lineages of the 1st wave of cell death. We have made the following textual change in the Result section on page 9:

Page 9

“Of note, psf-2(t3443ts) affects the death of all 1st wave cell deaths (Fig. 2A), which indicates that it does not cause a lineage-specific block in cell death.”

And then, from my understanding this along with the experiment on post-embryonic death does suggest that psf-2(ts) results in a “general” block, but it would help the reader to clarify perhaps the types of divisions of this first wave...e.g. are they all divisions that normally result in neuron and a cell death, or do these represent a variety of different fates vs cell death. Even most C. elegans readers won’t know which cell fates may be in these specific MS and AB sublineages.

Memar et al

As stated above, the surviving sisters of all 14 1st wave cell deaths do not differentiate but divide again and some of their daughters go through one more round of division.

4) Picky but important for accuracy: Several places it is stated that role of PSF-2 in cell death fate is independent or separate (several different wordings used) of its DNA helicase/unwinding activity. Because this isn’t being tested here in C. elegans, and it is unclear what this psf-2(ts) mutation does to the protein’s activity, I think it is more accurate to use wording that this role is separable from its role in DNA replication (and could add, thus presumably DNA helicase activity)

Memar et al

We thank the reviewer for pointing this out. We have made changes in seven relevant sentences throughout the manuscript:

Abstract, page 2

“We present evidence that this requirement is separable from the role of CMG in DNA replication.”

Introduction, page 4, 5

“Here we demonstrate that the increase in egl-1 BH3-only expression in the daughters that die is dependent on the eukaryotic replicative helicase CMG (Cdc45-MCM2-7-GINS), and we provide evidence that this requirement of CMG is separable from its role in DNA replication.”

Results, page 13

“The lack of correlation between the increased cell cycle length phenotype and the Ced phenotype observed in psf-2(t3443ts) mutants suggests that the role of PSF-2 GINS2 in the acquisition of the cell death fate is separable from its role in DNA replication and thus presumably its DNA helicase activity.”

Results, page 14

“Therefore, the role of psf-2 GINS2 in the acquisition of the cell death fate is separable from its role in DNA replication and thus presumably DNA helicase activity.”

Discussion, page 23 (changes in three separate sentences)

“Although coupled to replication, the function of Mcm2 in the differentiation of mouse ESCs in vitro is separable from its helicase activity and hence, independent of its well described role in DNA replication⁴⁸⁻⁵⁰. Similarly, we present in vivo evidence in support of the notion that the role of C. elegans CMG in cell fate divergence is separable from its role in DNA replication and thus presumably DNA helicase activity. Specifically, (1) increases in cell cycle length in psf-2(t3443ts) mutants (likely reflecting replication stress) do not correlate with the loss of the cell death fate and (2) increases in cell cycle length in tyms-1(e2300ts) mutants (also likely reflecting replication stress) do not lead to the loss of the cell death fate. Together, these findings suggest that in the context of asymmetric cell division, CMG has a conserved role in the divergence of cell fates that is distinct from its conserved role in DNA replication.”

Minor issues/suggestions to improve clarity or readability

Line 47 - “whether cell division itself...” I don’t think a good choice of words because that encompasses asymmetric segregation of components, mitotic spindle control etc and I don’t think what the authors intend

Memar et al

In response to the Reviewer’s comment, we have changed the sentences and paragraph in the Introduction in the following way:

Page 3

“A large body of work has demonstrated that different daughter cell fates can be established through the asymmetric inheritance – in the form of protein or mRNA - of cell fate determinants.

Whether a cell fate determinant can also be asymmetrically inherited in the form of the gene that encodes it is not clear. The quantal cell cycle theory proposes that changes at the chromosomal level during chromosome replication make regions of the genome available for transcription in daughter cell that were not available for transcription in the mother cell^{5,6}. Whether and how this may occur in vivo is unknown.”

Line 99 -For non C. elegans, even non-C elegans embryo experts, it would be helpful to give the reader an idea of the normal number of cells at the pre-morphogenetic stage, for comparison later when psf-2(RNAi) causes arrest at 50-cell stage (I know you didn't count, but you could state the wt # at this stage),.

Memar et al

In response to the reviewer's comments, we have added the following information regarding the pre-morphogenetic stage:

Page 6

“At the non-permissive temperature (25°C), the morphology of early t3443ts embryos is essentially indistinguishable from that of wild-type embryos (Fig. 1A, 4-cell stage, Pre-morphogenetic stage [~350-cell stage]) (see Materials and Methods for the exact time of the shift from permissive to non-permissive temperature).”

Line 132 section -Why this lineage? Overall the paper is very clearly written and flows nicely. However, since the abstract and Intro are about ACD and cell death, but it takes a long time to get to anything to do with those topics, I found myself wondering how you found this mutant in the first place. Were you examining lineage delays, or looking for new cell death mutants? And then specifically here starting with line 132, does this lineage have cell death? I didn't see any X's in the figure so assuming not and they appear different sublineages than in Fig. 2. Again, especially for a non-C. elegans person all the cell labels may be daunting and so stating more explicitly why these cells were examined could help with readability.

Memar et al

Concerning the reviewer's question why we chose to analyse the ABarp lineage (Figure 1D), we thank the reviewer for pointing out that we had failed to provide a rationale. The reviewer is correct, the ABarp lineage does not result in a cell death. In Figure 2 and 3, we mainly analysed AB-derived 1st wave cell deaths, which are defined as AB-derived cell deaths that occur after the 9th round of cell division. The cell ABarpppppp, derived from ABarp, is one of the few cells that differentiates after the 9th round of division (it differentiates into the hypodermal cell V6R). For this reason, the ABarp lineage and ABarpppppp represent appropriate controls for the lineaging data provided for cell-death lineages and cell deaths shown in Figures 2 and 3. To clarify this, we have modified the section introducing the ABarp lineage in the Result section on page 7:

Page 7

“We allowed one-cell embryos to undergo four rounds of cell division, identified the ABarp blastomere and measured cell cycle length during the five consecutive rounds of cell division that give rise to the cell ABarpppppp, which differentiates into the hypodermal cell V6R (Fig. 1C, D). ABarpppppp is one of the few cells that differentiates after the 9th round of cell division. For this reason, it serves as a control for 1st wave cell deaths, which occur after the 9th round of cell division.”

The reviewer also asks how we found the *psf-2(t3443ts)* mutant in the first place. We found the mutant in a screen for temperature-sensitive embryonic lethal mutants with abnormalities in the pattern of cell death and this was stated in the first sentence of the Result section on page 6:

Page 6

“We identified the *t3443ts* mutation by screening a collection of temperature-sensitive (ts) embryonic lethal mutants for abnormalities in the invariant pattern of cell death.”

Line 148 – It would help if you emphasized that you did longer RNAi of 48hr, which gave a stronger phenotype of arrest of 50cell. This is indicated in parentheses, but I had to go back and hunt for it and so missed the point the first time.

Memar et al

In response to the Reviewer’s comment, we have added the following two statements concerning *psf-2(RNAi)*:

Page 7

“(psf-2(RNAi) for 24h results in a weak loss-of-function phenotype.)”

Page 8

“(psf-2(RNAi) for 48h causes a strong loss-of-function phenotype.)”

Line186 – “This indicates that in 38% of...” The sentence does not follow from the data provided in the prior sentence or easily from the graph. An average time was given, not the number of cells that divided within the normal wt range vs out of that range, which is what the sentence seems to be indicating.

Memar et al

We thank the reviewer for pointing this out. In response, we have changed the relevant sentence in the following way:

Page 10

“This indicates that in the 1st wave cell deaths that still occur in psf-2(t3443ts) animals (38% of all 1st wave cell deaths analyzed), cell death is delayed.”

Changes made to figures

For transparency, below, we are listing the changes made to figures.

Figure 1B and new **Figure S5** (previous Figure S4): According to Nature Communication guidelines, we now indicate the individual worms analysed.

Figure 2: In **Figure 2a** in *psf-2(t3443ts)* Embryo #2, we obtained data for one more cell, and this cell inappropriately survived. This changes the n's analysed, as indicated also in the **Figure 2b** below.

In **Figure 2B**, the n's increased for some genotypes: *psf-2(t3443ts)*, *psf-2(t3443ts)* 15C, *psf-2(t3443ts); psf-2(+)*, *psf-3(RNAi)*, *mcm-2(RNAi)*. For some genotypes, this has consequences

for “% cell deaths blocked during the first wave”. The biggest change is for *mcm-7*(RNAi). We must have previously miscalculated “% cell deaths blocked during the first wave”. The value changed from 22% to 54%.

Genotype	% cell deaths blocked during first wave	n
+/+	0	52
ced-3 (n717)	100	32
psf-2 (t3443ts)	62	40
psf-2 (t3443ts) 15°C	0	26
psf-2 (t3443ts); psf-2 (+)	0	50
psf-2 (RNAi)	64	28
psf-3 (RNAi)	62	24
mcm-7 (RNAi)	22	26
mcm-2 (RNAi)	55	18
tyms-1 (e2300ts)	0	31

old version

Genotype	% cell deaths blocked during first wave	n
+/+	0	52
ced-3 (n717)	100	32
psf-2 (t3443ts)	63	41
psf-2 (t3443ts) 15°C	0	32
psf-2 (t3443ts); psf-2 (+)	0	51
psf-2 (RNAi)	64	28
psf-3 (RNAi)	60	25
mcm-7 (RNAi)	54	26
mcm-2 (RNAi)	58	19
tyms-1 (e2300ts)	0	31

new version

Figure 3: According to the Nature Communication guidelines, we are now indicating the p values directly in the figure. This changes **Figure 3B** as follows:

REVIEWERS' COMMENTS

Reviewer #1 (Remarks to the Author):

I have read the authors' responses to the comments of the reviewers. While they acknowledge the possibility of the helicase-transcription model I proposed, they do not perform any of the studies I suggested. I would recommend, therefore, that the authors at least acknowledge in the manuscript the possibility of this alternative model. They have already changed some of the language regarding the separate replication and cell death mechanisms in response to the other reviewer's comments, and this specific model could also be added to those changes. Otherwise, I have no other concerns.

Memar et al

To address Reviewer #1's comment regarding the helicase-transcription model, we have added the following statement in the Discussion:

"We acknowledge that it does remain formally possible that CMG helicase activity is required to remove torsional tension during transcription and that this hypothetical role accounts for the effects of CMG on gene expression."

Reviewer #2 (Remarks to the Author):

The authors have very nicely addressed all my major and minor comments. I have no further concerns.